# Neurexins in serotonergic neurons regulate neuronal survival, serotonin transmission, and complex mouse behaviors

Amy Cheung[1,2,3], Kotaro Konno[4], Yuka Imamura[5], Aya Matsui[6], Manabu Abe[7], Kenji Sakimura[7], Toshikuni Sasaoka[8], Takeshi Uemura[9,10], Masahiko Watanabe[4], Kensuke Futai[1,2]*

[1]Department of Neurobiology, University of Massachusetts Chan Medical School, Worcester, United States; [2]Brudnick Neuropsychiatric Research Institute, University of Massachusetts, Worcester, United States; [3]Medical Scientist Training Program, University of Massachusetts, Worcester, United States; [4]Department of Anatomy, Faculty of Medicine, Hokkaido University, Sapporo, Japan; [5]Departments of Pharmacology and Biochemistry & Molecular Biology, Institute for Personalized Medicine, Pennsylvania State University College of Medicine, 500 University Drive, Hershey, United States; [6]Vollum Institute, Oregon Health & Science University, Portland, United States; [7]Department of Animal Model Development, Brain Research Institute, Niigata University, Niigata, Japan; [8]Department of Comparative and Experimental Medicine, Brain Research Institute, Niigata University, Niigata, Japan; [9]Division of Gene Research, Research Center for Advanced Science, Shinshu University, Nagano, Japan; [10]Institute for Biomedical Sciences, Interdisciplinary Cluster for Cutting Edge Research, Shinshu University, Nagano, Japan

**\*For correspondence:**
kensuke.futai@umassmed.edu

**Competing interest:** The authors declare that no competing interests exist.

**Abstract** Extensive serotonin (5-hydroxytryptamine, 5-HT) innervation throughout the brain corroborates 5-HT's modulatory role in numerous cognitive activities. Volume transmission is the major mode for 5-HT transmission but mechanisms underlying 5-HT signaling are still largely unknown. Abnormal brain 5-HT levels and function have been implicated in autism spectrum disorder (ASD). Neurexin (*Nrxn*) genes encode presynaptic cell adhesion molecules important for the regulation of synaptic neurotransmitter release, notably glutamatergic and GABAergic transmission. Mutations in *Nrxn* genes are associated with neurodevelopmental disorders including ASD. However, the role of *Nrxn* genes in the 5-HT system is poorly understood. Here, we generated a mouse model with all three *Nrxn* genes disrupted specifically in 5-HT neurons to study how Nrxns affect 5-HT transmission. Loss of *Nrxns* in 5-HT neurons reduced the number of serotonin neurons in the early postnatal stage, impaired 5-HT release, and decreased 5-HT release sites and serotonin transporter expression. Furthermore, 5-HT neuron-specific *Nrxn* knockout reduced sociability and increased depressive-like behavior. Our results highlight functional roles for Nrxns in 5-HT neurotransmission, 5-HT neuron survival, and the execution of complex behaviors.

## Editor's evaluation

Neurexins control the assembly, maturation, and function of nerve cell synapses, and their genetic loss causes multiple neuropsychiatric diseases, including schizophrenia and autism spectrum disorder (ASD). This manuscript makes an important contribution, by showing convincingly that deletion of all

neurexins specifically in serotonergic neurons causes a defect in the survival of serotonergic neurons, in the establishment of serotonergic axonal inputs in various brain regions, in the generation of serotonin release sites, and in serotonin secretion in various brain regions, resulting in ASD-like and depression-related behavioral defects. Thus, not only fast-acting transmitter systems but also modulatory ones depend on neurexin function, and serotonergic signaling contributes to the clinical features of neuropsychiatric disorders caused by neurexin loss. These findings will be interesting to experts in basic neuroscience, psychiatry, and neurology alike.

## Introduction

Serotonin (5-hydroxytryptamine, 5-HT) neurons in the raphe nuclei (RN) project their axons throughout the brain and modulate social interactions, stress responses, and valence among other processes. Abnormalities in 5-HT signaling have been extensively reported in neuropsychiatric disorders including depression, anxiety disorders, schizophrenia (SCZ), and autism spectrum disorder (ASD) (*Lesch and Waider, 2012*). 5-HT reaches postsynaptic specializations through volume transmission or at synapses and synaptic triads (*Belmer et al., 2017*). While much work has focused on deciphering receptor and reuptake dynamics in 5-HT signaling, the functional component important for 5-HT release remains undefined.

Nrxn genes (*Nrxn1-3*) encode alpha-, beta-, and gamma- ($\alpha/\beta Nrxn1-3$, $\gamma Nrxn1$) isoforms, and regulate synapse specification and function (*Südhof, 2017*). Copy number variations and mutations in Nrxns are associated with ASD and SCZ (*Südhof, 2017*). Numerous studies of $\alpha$ and $\beta Nrxn$ KO mice demonstrate impaired excitatory and inhibitory synaptic transmission (*Südhof, 2017*). While Nrxns regulate fast synaptic transmission, no studies have examined the role of Nrxns in central neuromodulatory systems like the 5-HT system. Therefore, elucidating the impact of Nrxns in 5-HT transmission will allow a better understanding of pathophysiological mechanisms underlying neuropsychiatric disorders.

In this study, we investigated the functions of Nrxns in the 5-HT system by assessing signaling properties and behavior in 5-HT neuron-specific Nrxn triple knockout (TKO) mice. We demonstrated that the loss of *Nrxn* genes reduced 5-HT release and the number of 5-HT neurons and release sites in the mouse brain. Moreover, the lack of Nrxns in 5-HT neurons altered social behavior and depressive-like phenotypes. Our findings highlight Nrxns as functional regulators of 5-HT signaling and function.

## Results

### Expression of *Nrxn* isoforms in 5-HT neurons and validation of the Fev/RFP/NrxnTKO mouse line

To characterize *Nrxn* genes expressed in 5-HT neurons, we analyzed scRNAseq data from a published database consisting of over 900 single-cell 5-HT neuron datasets (~1 million reads/cell) which generated 11 different 5-HT neuron clusters from the principal dorsal raphe nucleus (DRN), caudal DRN (cDRN), and median raphe nucleus (MRN) (*Figure 1A*, *Figure 1—figure supplement 1*; *Ren et al., 2019*). Transcriptional expression of six *Nrxn* isoforms ($\alpha$-, $\beta Nrxn1/2/3$) in 11 clusters indicated that all 5-HT neuron clusters express at least one $\alpha$- and $\beta Nrxn$ isoform (*Figure 1B, C*). These results suggest that Nrxn proteins are expressed in 5-HT neurons in all RN regions.

To test the roles of Nrxns in 5-HT neurons, we generated 5-HT neuron-specific *Nrxn* TKO mice by crossing *Fev$^{Cre}$*, an ETS family transcription factor promoting 5-HT neuron-specific Cre expression, tdTomato (RFP) reporter, and triple Nrxn1/2/3 floxed lines (Fev/RFP/NrxnTKO) (*Scott et al., 2005*; *Uemura et al., 2022*). Cre-negative littermates and Fev/RFP mice were used as control (Cntl) cohorts. The specific deletion of Nrxns in 5-HT neurons was confirmed by single-cell RT-qPCR and -dPCR (*Figure 1D*, *Figure 1—figure supplement 1*). The Fev/RFP/NrxnTKO line was fertile and viable and did not demonstrate obvious differences in gross appearance.

### Reduced 5-HT release in Fev/RFP/NrxnTKO mice

While numerous studies indicate that Nrxns regulate fast neurotransmitter release including that of glutamate and GABA, no studies have tested Nrxn function in central neuromodulatory systems.

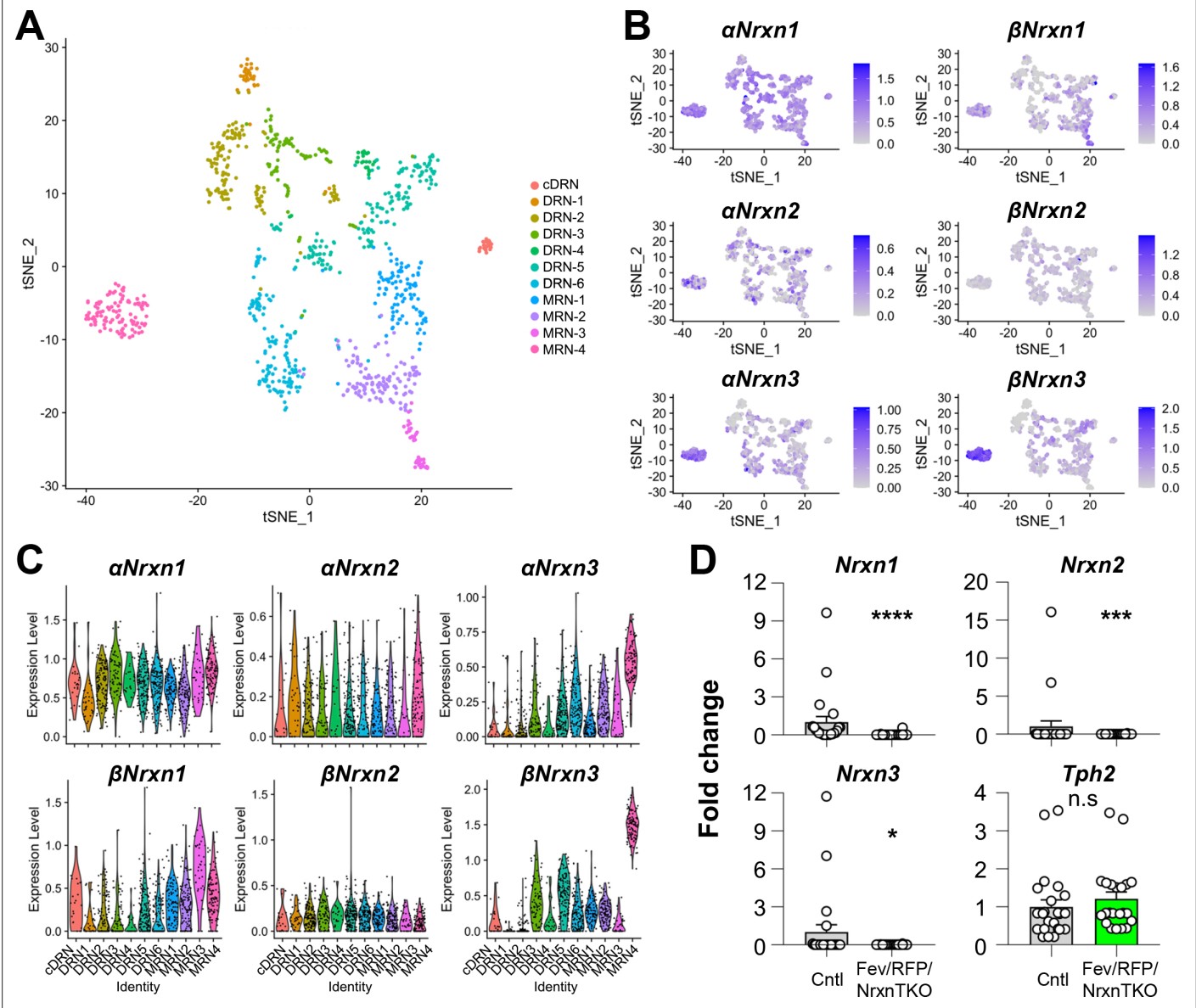

**Figure 1.** *Nrxn* expression in the raphe nuclei and confirmation of *Nrxn* deletion in Fev/RFP/NrxnTKO mice. Single-cell transcriptomic analysis of *Nrxn* expression in dorsal raphe nucleus (DRN), caudal DRN (cDRN), and median raphe nucleus (MRN) 5-hydroxytryptamine (5-HT) neurons. (**A**) Single-cell t-SNE plot of 999 Tph2-positive neurons representing 5-HT neurons analyzed from a recent publication (***Ren et al., 2019***). Eleven transcriptomic clusters were correlated with anatomical location. DRN, cDRN, and MRN 5-HT neurons have six, one, and four different subclusters, respectively. (**B**) Single-cell t-SNE plots for the expression of six *Nrxn* isoforms in distinct 5-HT neuron populations. Cells are colored by log-normalized expression of each transcript, and the color legend reflects the expression values of ln(CPM + 1). (**C**) Violin plots of six *Nrxn* splice isoforms in 11 clusters. Although there are cluster-dependent *Nrxn* expression patterns (e.g., α and β*Nrxn3* expression in MRN4), each *Nrxn* isoform is expressed in cells across all clusters. (**D**) Validation of the Fev/RFP/NrxnTKO mouse line. Expression of *Nrxn* genes in tdTomato-positive 5-HT neurons were compared between Fev/RFP (Cntl) and Fev/RFP/NrxnTKO mice. qPCRs against *Nrxn 1, 2, 3, Tph2,* and *Gapdh* (internal control) were performed for single-cell cDNA libraries prepared from tdTomato-positive neurons. Number of neurons: Fev/RFP (*n* = 23 neurons, 4 mice) and Fev/RFP/NrxnTKO (22, 4). Data are reported as mean ± standard error of the mean (SEM). n.s., not significant, *$p < 0.05$, ***$p < 0.001$, ****$p < 0.0001$; Mann–Whitney *U*-test.

The online version of this article includes the following source data, source code, and figure supplement(s) for figure 1:

**Source code 1.** Source code R file for t-distributed Stochastic Neighbor Embedding (tSNE) plots.

**Source data 1.** Source meta data for *Figure 1*.

**Figure supplement 1.** Validation of the Fev/RFP/NrxnTKO mouse line.

To directly examine the role of Nrxn on 5-HT release, we measured 5-HT transients in the DRN and hippocampus using fast-scan cyclic voltammetry (FSCV) (*Figure 2*). 5-HT release was recorded with a voltage ramp delivered through carbon-fiber electrodes (*Figure 2A*). Current–voltage (CV) plots displayed the expected currents for 5-HT oxidation and reduction peak potentials, indicating specificity for 5-HT (*Figure 2B*).

5-HT transients were recorded in the DRN where 5-HT neurons are highly clustered in littermate control and Fev/RFP/NrxnTKO mice (*Figure 2C–F*). Electrical stimulation was applied at two different stimulus strengths to evoke 5-HT release in acute brain slices containing the DRN. To confirm that electrically evoked FSCV transients were mediated by 5-HT release, the selective serotonin reuptake inhibitor fluoxetine (FLX) and action potential inhibiting sodium channel blocker tetrodotoxin (TTX) were applied (*Figure 2C, D, F*; *Carboni and Di Chiara, 1989*). Importantly, 5-HT peak amplitude was significantly reduced in Fev/RFP/NrxnTKO mice (*Figure 2E*). FLX caused a similar increase in 5-HT transient area in each genotype, indicating that Nrxn TKO did not change transporter activity (*Figure 2F*). Next, we performed FSCV recordings in the dorsal hippocampal CA3 region to determine whether differences in 5-HT release could be detected in a distal region receiving 5-HT fiber projections (*Figure 2G–J*). We observed robust suppression of 5-HT currents in Fev/RFP/NrxnTKO mice and no genotype-specific differences in response to FLX. Taken together, these findings indicate that Nrxns are important for 5-HT release.

## Reduced 5-HT fiber density and active zone number in Fev/RFP/NrxnTKO mice

Next, we analyzed whether Nrxns are important for 5-HT innervation in brain regions that receive 5-HT projections by analyzing RFP-positive fibers between Fev/RFP (Cntl) and Fev/RFP/NrxnTKO mice (*Figure 3A–I*; *Awasthi et al., 2021*). We found that RFP+ fibers were reduced in the hippocampus, DRN and MRN of Fev/RFP/NrxnTKO mice relative to controls. Interestingly, no differences were seen in the projections to the nucleus accumbens suggesting that 5-HT inputs are not globally altered. In addition, serotonin transporter (SERT) expression in the brain was also reduced in brainstem and hippocampus (*Figure 3—figure supplement 1*). These findings suggest that Nrxns selectively mediate 5-HT fiber area depending on the innervated circuit.

It has been reported that Nrxn TKO in cerebellar granule cells cause cell death (*Uemura et al., 2022*), which raises the possibility that the reduced 5-HT fiber density is due to 5-HT neuron cell death. To address this possibility, we compared the population of RFP-expressing 5-HT-positive 5-HT neurons in the DRN and MRN between Fev/RFP (Cntl) and Fev/RFP/NrxnTKO mice at different ages (*Figure 3L, M*). The RFP/5-HT ratio between Cntl and Fev/RFP/NrxnTKO mice was comparable at P7 but reduced by increasing age specifically in Fev/RFP/NrxnTKO mice suggesting the postnatal loss of Cre-positive 5-HT neurons in Fev/RFP/NrxnTKO mice. To further elucidate the mechanism underlying reduced 5-HT release in Fev/RFP/NrxnTKO mice, we confirmed the expression of RIM, a major active zone protein, in synaptophysin-positive 5-HT terminals in ultra-thin sections (*Figure 3—figure supplement 2*). Next, we performed confocal imaging comparing the expression of RIM1/2 in 5-HT fibers in the hippocampus between Cntl and Fev/RFP/NrxnTKO mice (*Figure 3N, O*). The density of RIM1/2 in Fev/RFP/NrxnTKO mice was lower than that in Cntl. These results suggest that Nrxn TKO reduces 5-HT release by decreasing the number of 5-HT neurons and release sites.

## Mild social behavior impairment in Fev/RFP/NrxnTKO mice

We investigated the behavior of adult Fev/RFP/NrxnTKO mice in a variety of assays. Basic activities, evaluated by locomotor activity, rotarod performance, and open field, did not differ between Fev/RFP/NrxnTKO mice and Cre-negative littermate controls (*Figure 4—figure supplement 1*). To examine the role of Nrxns in 5-HT system-related behaviors, we next assessed social behavior. Cntl and Fev/RFP/NrxnTKO underwent a direct social interaction test to examine naturally occurring interactions between a subject mouse and a juvenile stimulus mouse. In trial 1, the stimulus mouse was unfamiliar to the subject mouse. After 24 hr, the subject mouse was re-exposed to the same stimulus mouse (trial 2) (*Figure 4A*). Social investigation was measured across both trials and as a reduction in time that the subject mouse spent investigating the stimulus mouse in trial 2. We found that Fev/RFP/NrxnTKO mice spent less time exploring the stimulus mouse in trial 1 (*Figure 4B*) and differed in their investigation of the stimulus mouse across trials (*Figure 4C*) compared with littermate Cntl mice. These results

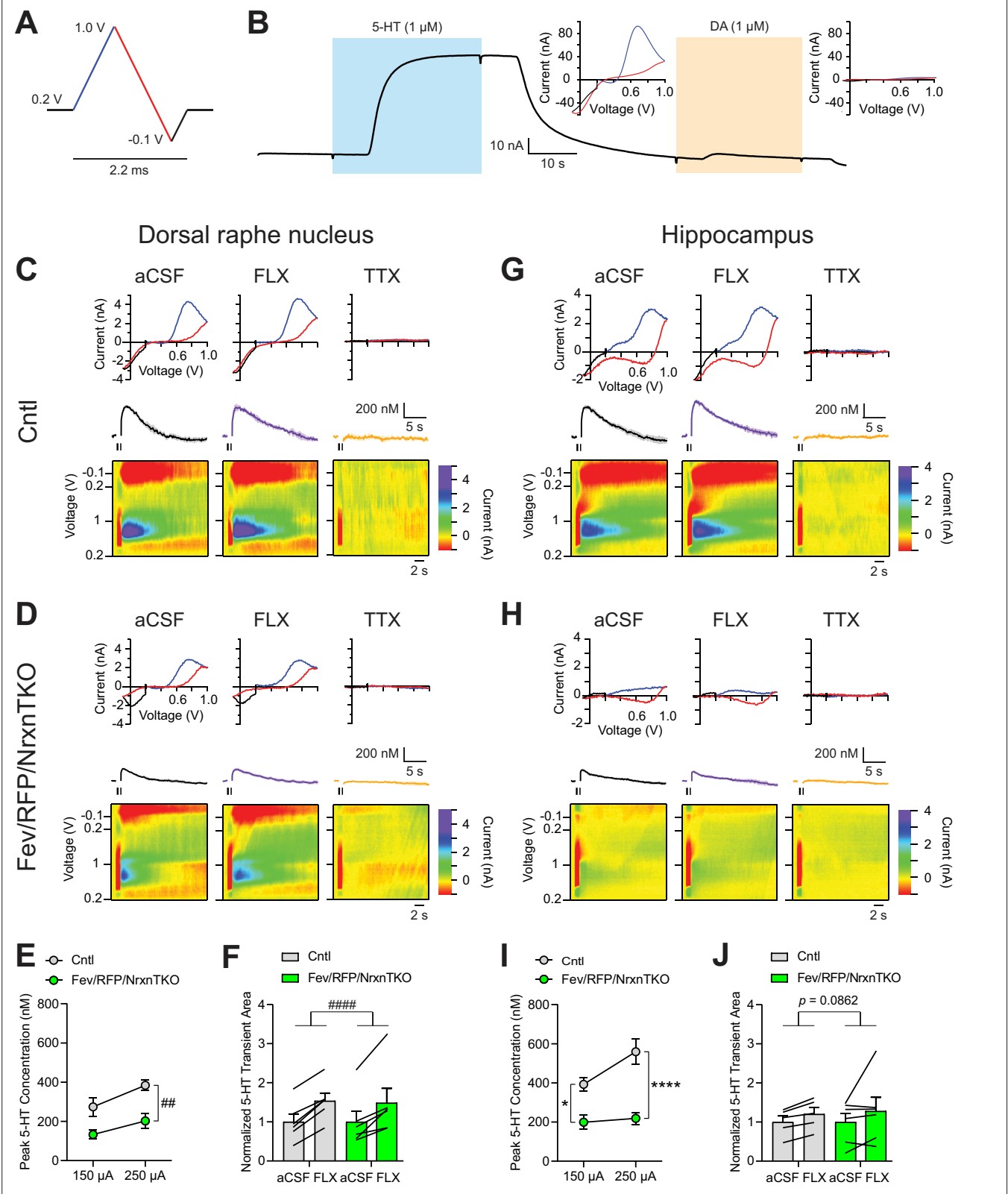

**Figure 2.** The lack of Nrxns in 5-hydroxytryptamine (5-HT) neurons reduces evoked 5-HT release in the dorsal raphe nucleus (DRN) and hippocampus. (**A**) Voltage ramp protocol for detecting 5-HT with fast-scan cyclic voltammetry (FSCV). Rapid cycling (2.2 ms) of the voltage ramp produced a current based on voltage-dependent oxidation (blue, oxi) and reduction (red) representing real-time 5-HT release. (**B**) Representative current traces when measuring 5-HT (1 µM, blue) or dopamine (DA; 1 µM, orange) using carbon-fiber microelectrodes calibrated on a 5-HT voltage ramp. Insets:

*Figure 2 continued on next page*

*Figure 2 continued*

Background-subtracted current–voltage (CV) plots of the electrochemical current when 5-HT (left) or DA (right) was applied. Electrically evoked 5-HT transients detected in the DRN of littermate control (Cntl) (n = 6 slices, 3 mice) (**C**) and Fev/RFP/NrxnTKO (6, 3) (**D**) mice during perfusion with artificial cerebrospinal fluid (aCSF) (left), serotonin transporter (SERT) blocker fluoxetine (FLX, 10 μM, middle), and action potential-blocking tetrodotoxin (TTX; 1 μM, right). Top: representative CV plots of the electrochemical current in aCSF, FLX, and TTX. Middle: average 5-HT transients under each condition. Bottom: background-subtracted 3D voltammograms (false color scale) as a function of time (x-axis, 20 s) and voltage applied (y-axis). Note that TTX completely abolished peak transients indicating that FSCV measurements were mediated by action potential-induced 5-HT release. The expected oxidation peaks for 5-HT in the CV plots acquired from Cntl and Fev/RFP/NrxnTKO mice were similar. FLX application prolonged 5-HT transient area which verified that the FSCV transients detected 5-HT signals. Cntl and Fev/RFP/NrxnTKO mice showed similar responses to FLX. (**E**) Peak amplitude of 5-HT transients evoked using a 150- or 250-μA stimulation train (30 pulses, 30 Hz, 1 ms) in the DRN of Cntl (gray) and Fev/RFP/NrxnTKO (green) mice. 5-HT release was significantly different between genotypes (two-way repeated measures analysis of variance [ANOVA]: genotype main effect, $F_{1,10} = 12.02$, ##p = 0.006; stimulation strength main effect, $F_{1,10} = 26.89$, p = 0.0004; stimulus strength × genotype interaction, $F_{1,10} = 1.936$, p = 0.2648). (**F**) Normalized area of 5-HT transients recorded in the DRN before and after FLX application. 5-HT transient area before and after FLX was significantly different (two-way repeated measures ANOVA: drug main effect, $F_{1,10} = 63.88$, ####p < 0.0001; genotype main effect, $F_{1,10} = 0.004378$, p = 0.9486; drug × genotype interaction, $F_{1,10} = 0.1492$, p = 0.7074). Electrically evoked 5-HT transients detected in the hippocampus of Cntl (n = 3) (**G**) and Fev/RFP/NrxnTKO (n = 3) (**H**) mice during perfusion with aCSF (left), SERT blocker FLX (10 μM, middle), and action potential-blocking TTX (1 μM, right). Top: representative CV plots of the electrochemical current in aCSF, FLX, and TTX. Middle: average 5-HT transients under each condition. Bottom: background-subtracted 3D voltammograms (false color scale) as a function of time (x-axis, 20 s) and voltage applied (y-axis). (**I**) Peak amplitude of 5-HT transients evoked using a 150- or 250-μA stimulation train in the hippocampus of Cntl and Fev/RFP/NrxnTKO mice. 5-HT release at each stimulation strength was significantly different between groups (two-way repeated measures ANOVA with Šidák's post hoc test following significant stimulation strength × genotype interaction, $F_{1,9} = 6.982$, p = 0.0268; stimulation strength main effect, $F_{1,9} = 11.24$, p = 0.0085; genotype main effect, $F_{1,9} = 24.56$, p = 0.0008). (**J**) Normalized area of 5-HT transients recorded in the hippocampus before and after FLX application. 5-HT transient area before and after FLX differed in Cntl mice (two-way repeated measures ANOVA: drug main effect, $F_{1,9} = 3.711$, p = 0.0862 genotype main effect, $F_{1,9} = 0.01351$, p = 0.91; drug × genotype interaction, $F_{1,9} = 0.09308$, p = 0.7672). n.s., not significant, *p < 0.05, ****p < 0.0001; Cntl vs. Fev/RFP/NrxnTKO: ##p < 0.01; aCSF vs. FLX: ####p < 0.0001.

The online version of this article includes the following source data for figure 2:

**Source data 1.** Source data for fast-scan cyclic voltammetry (FSCV) plots in *Figure 2E, F, I, J*.

suggest that Fev/RFP/NrxnTKO mice have deficits in sociability. To further study sociability in the Fev/RFP/NrxnTKO mice, we performed a three-chamber social interaction test (3CST), in which a stimulus mouse was confined to a cylinder to limit direct interaction (*Figure 4—figure supplement 2A*). Both littermate Cntl and Fev/RFP/NrxnTKO mice showed similar preferences for a stimulus mouse than for a second empty cylinder in a different chamber (days 1 and 2) and for a novel stimulus mouse than for the previously encountered juvenile conspecific (day 3) (*Figure 4—figure supplement 2B, C*). Cntl and Fev/RFP/NrxnTKO mice showed different investigation behavior on day 1. However, there were no differences in exploration time between groups which contrasts the sociability deficits observed in the direct social interaction test. Interestingly, one of the depression tests, the forced swim test but not tail suspension test, revealed increased immobility behavior in Fev/RFP/NrxnTKO compared with littermate Cntl mice (*Figure 4D, E*). Other tests addressing learning and memory and repetitive behaviors displayed no abnormalities in Fev/RFP/NrxnTKO mice (*Figure 4—figure supplement 3*). These results demonstrate that the absence of Nrxns in 5-HT neurons impairs direct social behavior and moderately influences depression-related behavior.

## Discussion

Nrxns regulate the release of fast neurotransmitters such as glutamate and GABA by coupling $Ca^{2+}$ channels to presynaptic release machinery (*Südhof, 2017*). The study of cerebellins, secreted protein-binding partners of Nrxns, indirectly implicates Nrxns in serotonergic system development (*Seigneur et al., 2021*). While cerebellin knockout was shown to alter DRN 5-HTergic circuits, Nrxn roles in central neuromodulatory systems have never been addressed. Here, we provide evidence that Nrxns control neuromodulatory 5-HT release. We found that the DRN and hippocampus displayed >40% reduction in 5-HT release in Fev/RFP/NrxnTKO mice. Fev expression begins in the embryonic stage and is found in a subpopulation of 5-HT neurons (~60% of 5-HT neurons) (*Scott et al., 2005*; *Figure 3M*). Therefore, the impact of Nrxn TKO during development and non-Cre expressing 5-HT neurons should be noted. However, consistent with the robust functional deficit observed in Fev/RFP/NrxnTKO mice, the structural deficit identified by RFP-positive fiber and release site densities were significant (>50%),

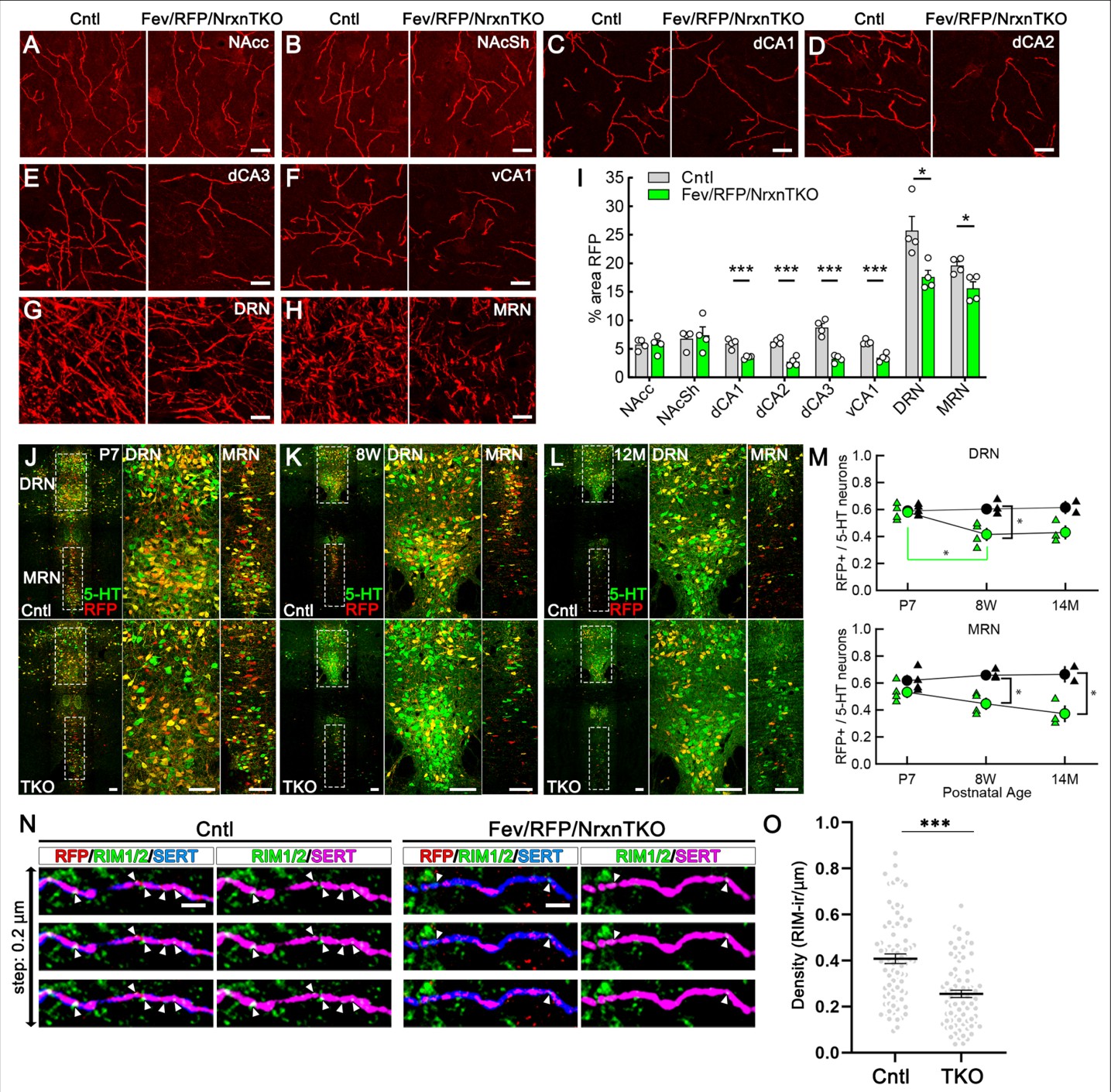

**Figure 3.** The absence of Nrxns in 5-hydroxytryptamine (5-HT) neurons decreases 5-HT fiber density, neuron number, and release sites. Representative ×100 images of RFP-positive fibers in the nucleus accumbens core (NAcc) (**A**), nucleus accumbens shell (NAcSh) (**B**), dorsal hippocampal CA1–3 subregions (dCA1, 2, 3) (**C–E**), ventral hippocampal CA1 (vCA1) (**F**), dorsal raphe nucleus (DRN) (**G**), and median raphe nucleus (MRN) (**H**). (**I**) Quantification of the area of RFP-positive fibers revealed that 5-HT innervation was altered in specific brain regions in mice lacking Nrxns ($n$ = 4 mice/ genotype; for each mouse, 6 fields of view were averaged for each region). *$p < 0.05$, ***$p < 0.001$; unpaired two-tailed Student's $t$-test. Scale bars, 10 μm. (**J–M**) Relative proportion of RFP-expressing 5-HT neurons in the DRN and MRN between Fev/RFP (Cntl, top row) and Fev/RFP/NrxnTKO (triple knockout, TKO, bottom row) mice at different ages: postnatal day 7 (P7) (**J**), 8 weeks (**K**), and >14 months old (**L**). (**M**) Quantification of RFP+ neurons as a fraction of 5-HT+ (green) neurons at three different postnatal ages. Note that the RFP+/5-HT ratio in TKO mice decreased with aging, suggesting that TKO in 5-HT neurons causes postnatal cell death. The numbers of Cntl and TKO mice were (postnatal age, number of mice): Cntl: P7 and 8 w, 4 and 12– 13 M, 2; TKO: P7 and 8 w, 4 and 12–13 M, 3. *$p < 0.05$, two-way analysis of variance (ANOVA) (**N, O**). (**N**) Three consecutive triple immunofluorescence

*Figure 3 continued on next page*

*Figure 3 continued*

images with 200 nm step for RFP, RIM1/2, and serotonin transporter (SERT) obtained from Fev/RFP (Cntl, left) and TKO (right) brains. (**O**) Summary of RIM1/2 signal density between Cntl and TKO hippocampal CA3 region. ***p < 0.0001; unpaired two-tailed Student's *t*-test. Scale bars, 10 µm (**A–H**), 100 µm (**J–L**), and 2 µm (**N**).

The online version of this article includes the following source data and figure supplement(s) for figure 3:

**Source data 1.** Source data for plots in *Figure 3I and M*, and *Figure 3—figure supplement 1*.

**Figure supplement 1.** The absence of Nrxns in 5-hydroxytryptamine (5-HT) neurons reduces serotonin transporter (SERT) expression.

**Figure supplement 1—source data 1.** Source data blots.

**Figure supplement 1—source data 2.** Unedited blotting images for brainstem and hippocampal (Hip) P2 proteins.

**Figure supplement 2.** The expression of RIM1/2 in 5-hydroxytryptamine (5-HT) fibers in 100 nm ultra-thin sections.

suggesting that the roles of Nrxns in the 5-HT system are cell survival and the formation of functional components important for 5-HT release.

Given the predominance of non-junctional specializations, we speculate that Nrxns reside at 5-HT release sites that lack a direct postsynaptic target. The ability of Nrxns to couple with release machinery triggering 5-HT vesicle exocytosis and their roles in postsynaptic differentiation at synapses are yet to be explored. Decreased 5-HT neuron number and release sites suggest that 5-HTergic Nrxns contribute to cell survival and fiber formation. Indeed, Nrxns are important for cerebellar granule cell survival through regulating the autocrine neurotrophic-factor (NTF) secretory machinery (*Uemura et al., 2022*). Sparse pan-Nrxn deletion has also been shown to blunt inferior olive neuron climbing fiber projections in the cerebellum while complete removal of Nrxns at climbing fiber synapses did not alter climbing fiber axons but impaired synaptic transmission (*Chen et al., 2017*). It is interesting that the loss of Cre+ 5-HT Nrxn TKO neurons is limited to the early postnatal stage. This may suggest that matured 5-HT neurons use the NTF supply from surrounding non-5-HT and Cre-negative 5-HT neurons. The possibility of Cre expression altering properties related to 5-HT neuron function in place of Nrxn deletion cannot be excluded. Further investigation is essential to understand the molecular mechanisms underlying cell death in Nrxn TKO 5-HT neurons.

Overall behavioral phenotypes reminiscent of ASD in Fev/RFP/NrxnTKO mice are milder than null Nrxn KO mouse lines, suggesting that Nrxns in other cell types are also important in complex behaviors (*Born et al., 2015*; *Dachtler et al., 2014*; *Etherton et al., 2009*; *Grayton et al., 2013*). The observed deficits in sociability in the direct social interaction test and in the depressive-related forced swim test contrast the normal behaviors of Fev/RFP/NrxnTKO mice in the 3CST and tail suspension tests. The 3CST limits direct interaction with a stimulus mouse and it is possible that the mode and novelty, as there were genotype differences observed on initial encounter, of social interaction are critical to Fev/RFP/NrxnTKO mice. Additionally, the forced swim test was performed following the tail suspension test, and it is possible that Fev/RFP/NrxnTKO mice are more susceptible to stress rather than despair-associated coping responses. Of note, the *Fev^Cre* line limits Cre expression in up to 60% of 5-HT neurons (*Figure 3M*), indicating a need for a more 5-HT neuron-specific Cre line to fully understand the roles of 5-HTergic Nrxns in animal behaviors.

Our results reveal that Nrxns expressed in midbrain 5-HT neurons are important for cell survival and maintaining the presynaptic molecular function of 5-HT release sites. Further studies are necessary to decipher Nrxn-mediated 5-HT release machinery, examine the consequences of Nrxn deletion in RN-innervated circuits in other brain regions, and address whether 5-HT therapeutics can improve behavioral deficits.

## Materials and methods
### Animals

All experiments were conducted under approved animal protocols from the Institutional Animal Care and Use Committee (IACUC) at the University of Massachusetts Chan Medical School. 5-HT neuron-specific tdTomato mice (Fev/RFP) were generated by crossing Rosa26 with *LSL-tdTomato* (*^lox-STOP-lox^TdTomato*, Ai9 line: Jax #007905) and *Fev^cre* mice (ePet^Cre^: Jax #012712) (*Scott et al., 2005*). Both *Fev^Cre* and RFP lines were backcrossed with C57BL/6J line for at least 10 generations. Fev/RFP mice were crossed with Nrxn1^f/f^/2 ^f/f^/3 ^f/f^ mouse line (*Uchigashima et al., 2020a*; *Uemura et al.,*

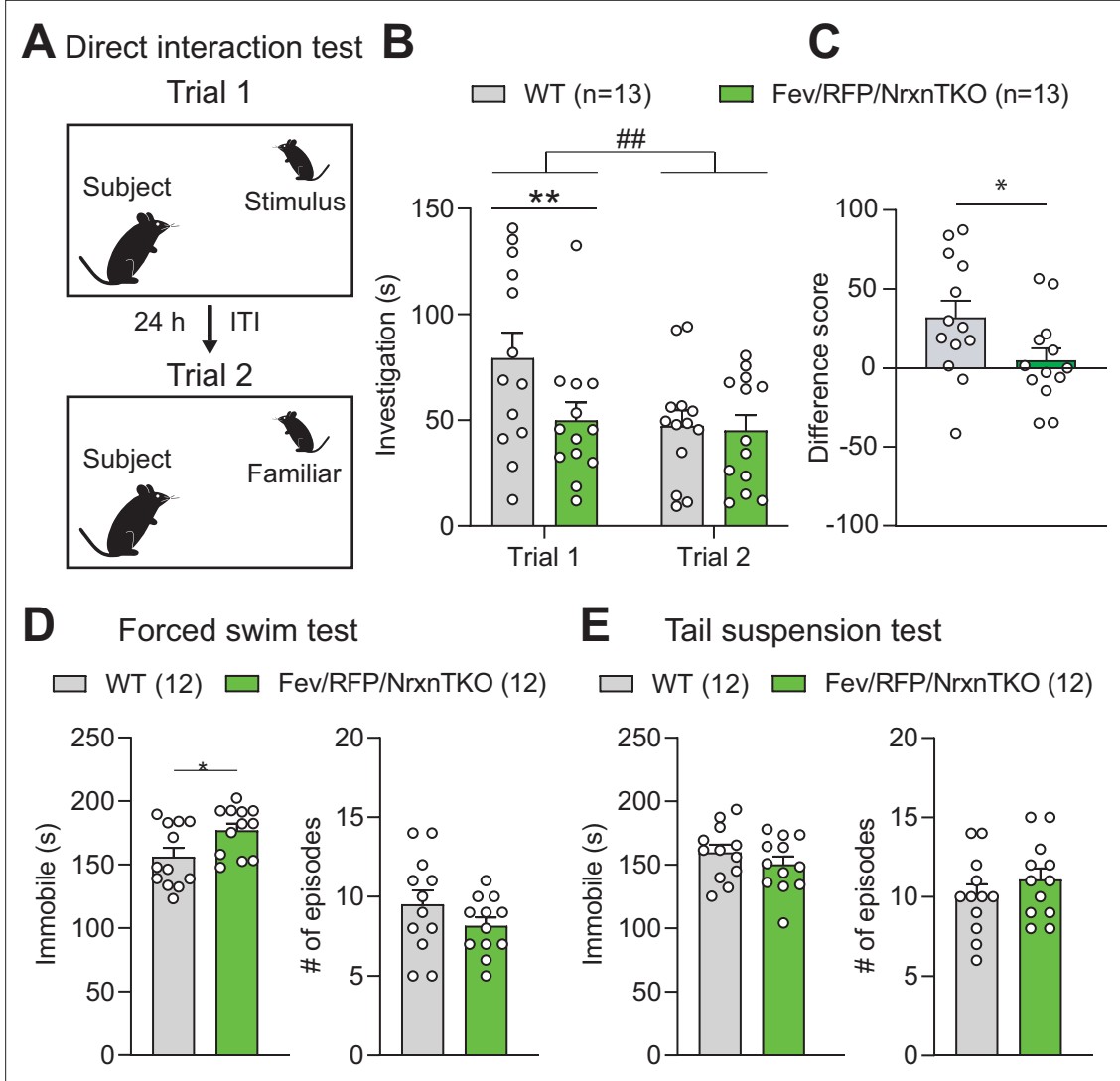

**Figure 4.** The absence of Nrxns in 5-hydroxytryptamine (5-HT) neurons impairs social behavior. (**A**) Direct social interaction test using the same juvenile stimulus across two trials. (**B**) Littermate control (Cntl) (gray, $n$ = 13) and Fev/RFP/NrxnTKO (green, $n$ = 13) mice differed in their investigation of the juvenile stimulus across the two trials. Both groups spent less time exploring the juvenile stimulus during trial 2 than in trial 1 (two-way repeated measures analysis of variance [ANOVA]: trial main effect, $F_{1,24}$ = 7.855, ##p = 0.0099; genotype main effect, $F_{1,24}$ = 2.086, p = 0.1616; significant trial × genotype interaction, $F_{1,24}$ = 4.344, p = 0.0479). Šidák's post hoc test identified a significant genotype difference in investigation time in trial 1 (**p = 0.0041). (**C**) The difference score of the interaction time across trials was reduced in Fev/RFPNrxnTKO mice (unpaired two-tailed Student's t-test: $t_{24}$ = 2.084, *p = 0.0479). (**D**) *Left*, Fev/RFP/NrxnTKO mice displayed increased immobile time compared with Cntl in the forced swim test (unpaired two-tailed Student's t-test: $t_{22}$ = 2.317, *p = 0.0302). *Right*, no difference in the number of immobile episodes was observed (unpaired two-tailed Student's t-test: $t_{22}$ = 1.301, p = 0.2068). (**E**) *Left*, Cntl ($n$ = 12) and Fev/RFP/NrxnTKO ($n$ = 12) mice showed no difference in time immobile in the tail suspension test (unpaired two-tailed Student's t-test: $t_{22}$ = 1.070, p = 0.2964). *Right*, there was no difference between genotypes in the number of immobile episodes (unpaired two-tailed Student's t-test: $t_{22}$ = 1.001, p = 0.3279).

The online version of this article includes the following source data and figure supplement(s) for figure 4:

**Source data 1.** Source data for *Figure 4*.

**Figure supplement 1.** 5-Hydroxytryptamine (5-HT) neuron-specific Nrxn TKO does not alter basic behavioral activities.

**Figure supplement 2.** Social behavior in the three-chamber social interaction test is preserved in Fev/RFP/NrxnTKO mice.

**Figure supplement 3.** 5-Hydroxytryptamine (5-HT) neuron-specific Nrxn TKO mice display normal learning and memory and repetitive behaviors.

*2022*) to generate 5-HT neuron-specific triple Nrxn knockout mouse line (Fev[Cre/lox-STOP-lox/lox-STOP-lox]tdTomato/Nrxn1[f/f]/2[f/f]/3[f/f]: Fev/RFP/NrxnTKO). The Fev/RFP/NrxnTKO line was maintained by breeding Fev/RFP/NrxnTKO mice with littermate Cre-negative ([lox-STOP-lox]tdTomato/Nrxn1[f/f]/2[f/f]/3[f/f]: Cntl) mice. Unless specified, Cre-negative littermates were used as controls. Male mice were used in all experiments. For social behavioral experiments, juvenile mice used as stimuli were 4- to 6-week-old male mice on a C57BL/6J background. Unless otherwise noted, 8- to 10-week-old mice were used for experiments.

Mice were group housed (2–5 per cage) and maintained in ventilated cages with ad libitum access to food and water on a standard 12-hr light/12-hr dark cycle (lights ON at 7 AM) in a temperature-controlled (20–23°C) facility. One to two weeks prior to experimentation, mice were acclimated to a reversed light/dark cycle (lights ON at 7 PM).

## Single-cell RNA extraction and RT-qPCR and -dPCR

All RT-qPCR and -dPCR experiments were performed on male mice aged 10 weeks or older. The whole procedure was done based on our recently developed protocol (*Mao et al., 2018*; *Uchigashima et al., 2020a*; *Uchigashima et al., 2020b*). Briefly, cytosol from RFP+ 5-HT neurons in the DRN and MRN were harvested from Fev/RFP (Cntl) and Fev/RFP/NrxnTKO mice using the whole-cell patch-clamp approach. A SMART-Seq HT Kit (TAKARA Bio) was used to prepare the amplified cDNA templates following the manufacturer's instructions (*Uchigashima et al., 2020a*; *Uchigashima et al., 2020b*). To assess Nrxn expression in individual 5-HT neurons of control and Fev/RFP/NrxnTKO mice, the following TaqMan gene expression assays (Applied Biosystems) with a FAM dye on their probes were used: *Nrxn1* (Mm03808857_m1), *Nrxn2* (Mm01236856_m1), *Nrxn3* (Mm00553213_m1), *Tph2* (Mm00557715_m1), and *Gapdh* (Mm99999915_g1). Additionally, a custom PrimeTime Std qPCR Assay, which provides the same type of quantitative PCR assay as TaqMan assays by utilizing hydrolysis probes in conjunction with gene-specific primer pairs, was designed for the TATA-box-binding protein (TBP) housekeeping gene with a HEX (VIC) dye on its probe (Integrated DNA Technologies, Inc). The assay consisted of a forward primer (5'-GGGAGAATCATGGACCAGAACA-3'), a reverse primer (5'-GGTGTTCTGAATAGGCTGTGG-3'), and a probe (/5HEX/CCTTCCACC/Zen/T TATG CTCAGGGC TT/3IABkFQ/). All the PCR reactions and analyses were performed blind to genotype. For the real-time quantitative PCR (qPCR) analysis, StepOnePlus qPCR system (Applied Biosystems) and the relative expression of *Nrxns* or *Tph2* were calculated as: Relative expression = $2^{Ct,Gapdh}/2^{Ct,Nrxns\ or\ Tph2}$; Ct, threshold cycle for target gene amplification, and presented as fold changes relative to that of Cntl. For the digital PCR (dPCR) analysis, a QuantStudio 3D Digital PCR System and its accompanying consumables were used and data were analyzed with QuantStudio 3D AnalysisSuite Cloud Software (Thermo Fisher). The absolute copy number of Nrxn2, Nrxn3, and TBP was measured and the relative abundance of Nrxn2 and Nrxn3 was calculated by normalizing their copy numbers to that of TBP.

## Transcriptome analysis

Single-cell RNA-seq data were obtained from a recent publication (*Ren et al., 2019*). See *Ren et al., 2019* for the single-cell isolation and sequencing.

**Table 1.** *Nrxn* transcript IDs used for quantification.

| ENSMUST00000072671.13 | *αNrxn1* |
| --- | --- |
| ENSMUST00000160844.9 | *αNrxn1* |
| ENSMUST00000174331.7 | *αNrxn1* |
| ENSMUST00000159778.7 | *βNrxn1* |
| ENSMUST00000174337.7 | *βNrxn1* |
| ENSMUST00000161402.9 | *αNrxn1* |
| ENSMUST00000054059.14 | *αNrxn1* |
| ENSMUST00000172466.7 | *βNrxn1* |
| ENSMUST00000160800.8 | *αNrxn1* |
| ENSMUST00001113462.7 | *αNrxn2* |
| ENSMUST00000236635.1 | *αNrxn2* |
| ENSMUST00001113461.7 | *αNrxn2* |
| ENSMUST00000235714.1 | *αNrxn2* |
| ENSMUST00001137166.7 | *αNrxn2* |
| ENSMUST00000167734.7 | *αNrxn3* |
| ENSMUST00000190626.6 | *αNrxn3* |
| ENSMUST00000167103.7 | *αNrxn3* |
| ENSMUST00000057634.13 | *αNrxn3* |
| ENSMUST00000238943.1 | *βNrxn3* |
| ENSMUST00000110133.8 | *βNrxn3* |
| ENSMUST00000110130.3 | *βNrxn3* |
| ENSMUST00000167887.7 | *αNrxn3* |

## Data processing and clustering

Datasets were downloaded from NCBI Gene Expression Omnibus (GSE135132). Reads were aligned to a mouse reference transcriptome (Mus_musculus.GRCm38.cdna.all.fa) using kallisto (*Bray et al., 2016*). Tximport R package (*Soneson et al., 2015*) was used to summarize the reads to the gene level. Each isoform was summarized manually to account for inclusion of spliced exons in the α or β Nrxn isoforms. The manually curated transcript IDs are provided in *Table 1*. Gene count data were analyzed using Seurat R package v4.0.1 (*Hao et al., 2021*). After excluding cells with low sequencing depth (50,000 reads) and low number of detected genes (cut-off was set at 7500 genes), the remaining 945 cells were assigned to clusters according to *Ren et al., 2019*. Counts were normalized for each cell using the natural logarithm of 1 + counts per 10,000 [ln(1 + counts/10k)]. Cells were visualized using a two-dimensional t-distributed Stochastic Neighbor Embedding (tSNE) and violin plots. The R code is provided as codeR (*Figure 1—source code 1*).

## Immunoblotting

Membrane fractions (P2 fraction) were used for immunoblottings. The whole experiments and analyses were performed blind to genotype. Adult mouse brains (8–10 weeks old) were homogenized in ice-cold buffer (5 mM 4-(2-hydroxyethyl)-1-piperazineethanesulfonic acid (HEPES) [pH 7.4], 1 mM $MgCl_2$, 0.5 mM $CaCl_2$, 1 mM NaF), and protease inhibitors (Roche cOmplete protease inhibitor: 05892970001) with a Teflon homogenizer (12 strokes). The homogenate extract was centrifuged at low speed (1400 × g for 10 min). The supernatant was centrifugated at 13,800 × g for 10 min. The supernatant was removed, and the P2 was resuspended in modified Radio Immunoprecipitation Assay (RIPA) buffer (50 mM Tris–HCl pH 8.0, 150 mM NaCl, 0.1% Triton X-100, 0.5% sodium deoxycholate, 0.1% sodium dodecyl sulfate (SDS), 1 mM sodium orthovanadate, 1 mM NaF, Protease inhibitor tablet). Homogenates (15 µg) were mixed with Sample Buffer (4% SDS, 20% glycerol, 0.004% bromophenol blue, 0.125 M Tris–Cl, pH 6.8, 2.5% 2-mercaptoethanol) to undergo SDS–polyacrylamide gel electrophoresis. Proteins were transferred to PVDF membranes (0.2 µm, Bio-Rad) and all remaining steps were performed at room temperature. Membranes were blocked in 5% skim milk and 5% bovine serum albumin in Tris-buffered saline (TBS) for 1 hr and then washed in TBS with 0.1% Tween 20 detergent (TBST) for 10 min. Membranes were incubated with primary antibodies prepared in 1% skim milk and 1% bovine serum albumin in TBST for 2 hr (mouse anti-SERT 1:7000, MAb Technologies, ST51-2; rabbit anti-Tph2 1:1000, Abcam, ab184505, RRID: AB_2892828; mouse anti-βActin 1:5000, Sigma, A1978, RRID: AB_476692). After washing with TBST, membranes were incubated with secondary antibodies for 1 hr (mouse and rabbit HRP, 1:1000, Millipore). Immunoblot signals were detected using an ECL detection kit (PerkinElmer Life Sciences) and the Bio-Rad Chemdoc system (Li-Cor). Quantification was performed by ImageJ software.

## Immunohistochemistry

All mice (8–10 weeks old) were transcardially perfused with ice-cold 4% paraformaldehyde (PFA)/0.1 M phosphate buffer (PB, pH 7.4) under isoflurane anesthesia. Brains were dissected and post-fixed at 4°C in PFA for 2 hr, then cryo-protected in 30% sucrose/0.1 M PB. Coronal 40-µm-thick brain sections were cut on a cryostat (CM3050 S, Leica Biosystems). All immunohistochemical incubations were carried out at room temperature. Sections were permeabilized for 10 min in 0.1% Tween 20/0.01 M phosphate-buffered saline (PBS, pH 7.4), blocked for 30 min in 10% normal donkey serum and incubated overnight in anti-SERT (guinea pig, 1 µg/ml, Frontier Institute, HTT-GP-Af1400, RRID: AB_2571777), anti-5-HT (goat, 1:1000, Abcam, ab66047, AB_1142794), anti-RIM1/2 (rabbit, 1:1000, Synaptic Systems, Rb 140203, AB_887775), or anti-RFP (Rabbit, 1:1000, Rockland, 21513) antibodies. The following day, sections were washed extensively then incubated in donkey anti-guinea pig-Alexa488, goat-Alexa488, and mouse-Alexa405 antibodies for 2 hr at a dilution of 1:500 (Jackson ImmunoResearch Laboratories). Sections were then mounted on slides (ProLong Gold, Invitrogen, P36930) and viewed for acquisition and analysis. Immunohistochemistry for neurotransmitter release machinery was confirmed using consecutive ultra-thin (100 nm) sections. Modified PFA (4% PFA and 0.1% glutaraldehyde in PB)-fixed brains were embedded in durcupan (Sigma) and consecutive ultra-thin sections were prepared by Ultracut ultramicrotome (Leica Microsystems). After etching with saturated sodium ethanolate solution for 1–5 s, ultra-thin sections on slides were treated with ImmunoSaver (Nisshin EM) at 95°C for 30 min. Sections were permeabilized for 10 min in 0.1% Triton X-100/0.01 M PBS (pH 7.4),

blocked for 30 min in 10% normal donkey serum, and incubated overnight in anti-SERT (goat, 1 µg/ml, Frontier Institute, MSFR103270, RRID: AB_2571776), RIM1/2 (Rabbit, Synaptic Systems, 140203, RRID: AB_887775, 1:1000), synaptophysin (guinea pig, 1 µg/ml, Frontier Institute, MSFR105690, RRID: AB_2571843), and anti-RFP (guinea pig, Frontier Institute, MSFR101410, RRID: AB_2571648) antibodies. The following day, sections were washed extensively then incubated with a mixture of Alexa 488-, Cy3-, or Alexa 647-labeled species-specific secondary antibodies for 2 hr at a dilution of 1:200 (Invitrogen; Jackson ImmunoResearch, West Grove, PA). Sections were then mounted on slides (ProLong Gold, Invitrogen, P36930) and viewed for acquisition and analysis.

## Imaging

Image acquisition and analysis were performed blind to genotype. RFP density analysis: We analyzed 5-HT fiber innervation to the nucleus accumbens core and shell (NAcc, NAcSh; Bregma 1.18 ± 0.3 mm), stratum oriens of the CA1, CA2, and CA3 subregions of the dorsal hippocampus (dCA1, dCA2, dCA3; Bregma −1.46 ± 0.4 mm), stratum oriens of the CA1 subregion of the ventral hippocampus (vCA1; −3.16 ± 0.4 mm), DRN (Bregma −4.56 ± 0.4 mm), and MRN (Bregma −4.5 ± 0.4 mm). To assess the density of RFP fiber inputs, four stained sections from each of four Fev/RFP (Cntl) and Fev/RFP/NrxnTKO brains containing the nucleus accumbens, hippocampus, or RN were imaged (1024 × 1024 pixels) using a laser scanning confocal microscope (LSM700, Zeiss) with a ×63 oil-immersion objective (NA 1.4) at an optical zoom of 1.6 and Zen black image acquisition software (Zeiss). For each brain, six randomly chosen ×100 fields of view within the region of interest were acquired with 21 z-stack steps at 0.35 µm spacing providing a 7-µm z-depth to generate maximum intensity projections (MIPs) of the z-stacks. Images from all brains for a particular region were acquired using identical settings, and data analyses were performed using ImageJ as previously described (*Werneburg et al., 2020*). The six images from each region per animal were averaged to generate a mean for that region in each animal, with *n* = 4 animals per genotype. A consistent threshold range was determined by subjecting images, blinded to genotype, to background subtraction and manual thresholding for each MIP within one experiment (IsoData segmentation method, 15–225). Using the analyze particles function, the thresholded images were used to calculate the total area of RFP fiber inputs.

### RFP-positive cell density analysis

We analyzed the relative expression of RFP- and 5-HT-positive neurons in the DRN (Bregma −4.6 ± 0.3 mm) and MRN (Bregma −4.5 ± 0.4 mm). The ratio was derived by dividing the number of RFP- and 5-HT-positive neurons (*n* = 2–5 mice per cohort). To assess the density of RFP-positive 5-HT neurons, three stained sections from each of four or five Fev/RFP (Cntl) and Fev/RFP/NrxnTKO brains containing the DRN or MRN were imaged (1024 × 1024 pixels) using a laser scanning confocal microscope (LSM700, Zeiss) with a ×20 water-immersion objective (NA 1.0) at an optical zoom of 0.5 and Zen black image acquisition software (Zeiss). For each section, two z-stack steps at 10–15 µm spacing were used to image the different 5-HT neuron populations. The ratios obtained from four to six images from each brain were averaged to generate a mean for the MRN and DRN of each animal. Images from the DRN and MRN were acquired using identical settings, and data analyses were performed using ImageJ as previously described (*Uchigashima et al., 2020a*).

### RIM1/2 density analysis

We analyzed RIM1/2 expression in 5-HT terminals projected in the CA1 pyramidal layer. To assess the density of RIM1/2 puncta in RFP- and SERT-positive fibers, four stained sections from each of four Fev/RFP (Cntl) and Fev/RFP/NrxnTKO brains containing the dorsal hippocampus were imaged (1024 × 1024 pixels) using a laser scanning confocal microscope (Olympus, FV1200) with a ×60 oil-immersion objective (NA 1.4). For each brain, 10–15 consecutive images were acquired with 0.2 µm steps and 18–33 RPF-/SERT-positive fibers were chosen. Because confocal microscopes can reach *z*-axis resolutions of 500 nm (*Schermelleh et al., 2010*), we measured the RIM signal found in two or more consecutive RFP-/SERT-positive terminals. RPF-/SERT-positive fibers in Cntl (81 fibers) and Fev/RFP/NrxnTKO (78) mice (*n* = 3 mice per genotype), respectively, were subjected to imaging analysis using MetaMorph software (Molecular Devices, Foster City, CA).

## Electrophysiology

### Slice preparation

All electrophysiological experiments were performed on male mice aged 10 weeks or older. Mice were anesthetized with isoflurane and decapitated. Brains were removed and quickly cooled in ice-cold, pre-oxygenated (95% $O_2$/5% $CO_2$) aCSF containing the following (in mM): 126 NaCl, 2.5 KCl, 1.2 $NaH_2PO_4$, 1.2 $MgCl_2$, 2.4 $CaCl_2$, 25 $NaHCO_3$, 20 HEPES, 11 D-glucose, 0.4 ascorbic acid, pH adjusted to 7.4 with NaOH. Coronal slices (400 μm) containing the dorsal hippocampus or DRN were prepared in ice-cold aCSF using a vibratome (VT1200 S, Leica Biosystems). Slices were recovered in oxygenated aCSF at room temperature (22–24°C) for at least 1 hr before use. Slices were then transferred to a recording chamber perfused at a rate of 1 ml/min with room temperature, oxygenated aCSF.

### Fast-scan cyclic voltammetry

5-HT measurements were performed in the radiatum of dorsal CA3 and DRN. All experiments and analyses were performed blind to genotype. To detect 5-HT release, carbon-fiber electrodes were prepared as previously described (*Hashemi et al., 2009*; *Matsui and Alvarez, 2018*). Carbon-fiber electrodes consisted of 7-μm diameter carbon fibers (Goodfellow) inserted into a glass pipette (A-M Systems, cat# 602500) with ~150–200 μm of exposed fiber. The exposed carbon fibers were soaked in isopropyl alcohol for 30 min to clean the surface. Next, the exposed fibers were coated with Nafion solution (Sigma) to improve detection sensitivity by inserting the carbon fiber into Nafion solution dropped in a 3-mm diameter circle of twisted reference Ag/AgCl wire for 30 s with constant application of +1.0 V potential. The carbon-fiber electrodes were air dried for 5 min and then placed in a 70°C oven for 10 min. A modified 5-HT voltage ramp was used, in which the carbon-fiber electrode was held at +0.2 V and scanned to +1.0 V, down to −0.1 V, and back to +0.2 V at 1000 V/s delivered every 100ms. Prior to recording, the electrodes were conditioned in aCSF with a voltage ramp delivered at 60 Hz for 10 min.

5-HT release was evoked with electrical stimulation (30 pulses, 30 Hz, 150 or 250 μA, 1 ms) from an adjacent custom-made bipolar tungsten electrode every 10 min. The stimulating electrode was placed ~100–200 μm away from the carbon-fiber electrode (*John et al., 2006*). Recordings were performed using a Chem-Clamp amplifier (Dagan Corporation) and Digidata 1550B after low-pass filter at 3 kHz and digitization at 100 kHz. Data were acquired using pClamp10 (Molecular Devices) and analyzed with custom written VIGOR software using Igor Pro 8 (32-bit; Wavemetrics) running mafPC (courtesy of M.A. Xu-Friedman). Carbon-fiber electrodes were calibrated with 1 μM 5-HT (Serotonin HCl, Sigma) at the end of the experiment to convert peak current amplitude of 5-HT transients to concentration. Three consecutive traces were averaged from each recording condition for analysis. Background-subtracted peak 5-HT transients and area under the curve were determined by subtracting the current remaining after TTX (tetrodotoxin citrate, Hello Bio) application from the maximum current measured. Dopamine HCl and FLX HCl were obtained from Sigma.

## Behavioral assays

All behavioral experiments were performed on male mice aged 8 weeks or older. Animals were habituated to the testing room for at least 30 min before each experiment and all tests were conducted under dim red-light conditions and white noise to maintain a constant ambient sound unless otherwise noted. All experiments and analyses were performed blind to genotype. Animals were used in only one behavioral paradigm for the direct social interaction test and fear conditioning. Mice underwent tests for locomotion, anxiety, repetitive behaviors, and depression in the following order: locomotor activity, open field, elevated plus-maze, grooming, marble burying, tail suspension test, and forced swim test. At least 2 days of rest were given in between all tests except for the tail suspension test and forced swim test, during which mice were allowed to rest for at least 7 days in between. Another cohort of mice completed the object interaction test followed by at least 2 days of rest before undergoing rotarod. Behavioral testing apparatuses were cleaned with 0.1% Micro-90 (International Products Corporation) between each mouse.

## Locomotor activity

Locomotor activity of each mouse was tracked in photobeam activity chambers (San Diego Instruments) for 90 min. Total horizontal movement was measured in 5-min bins.

## Rotarod

Motor coordination and balance were evaluated on a rotarod apparatus (San Diego Instruments) with an accelerating rotarod test. In each trial, mice were habituated to a rod rotating at 6 rpm for 30 s, then the rotation was increased to 60 rpm over 5 min. The latency to fall was measured over five trials with an interval of 10 min between each trial. Any mice that remained on the apparatus after 5 min were removed and their time was scored as 5 min.

## Open field

Mice were placed in the center of an open arena (41 × 38 × 30.5 cm) facing the furthest wall and allowed to freely explore the arena for 10 min. Time spent in the center of the arena (20.5 × 19 cm) was automatically tracked with EthoVision XT 11.5 (Noldus).

## Elevated plus-maze

The apparatus (Med Associates) consists of four arms, two enclosed with black walls (19 cm high) and two open (35 × 6 cm), connected by a central axis (6 × 6 cm) and elevated 74 cm above the floor. Mice were placed in the intersection of the maze facing the furthest open arm and allowed to freely explore the maze for 5 min. Time spent in the open and closed arms (index of anxiety-like behavior) and total entries into the open and closed arms (index of locomotor activity) were automatically measured with MED-PC IV software.

## Direct social interaction test

The test was adapted from *Hitti and Siegelbaum, 2014*. Each mouse was placed individually into a standard mouse cage and allowed to habituate for 5 min followed by the introduction of a novel male juvenile mouse. The activity was monitored for 10 min and social behavior initiated by the subject mouse was measured by an experimenter sitting approximately 2 m from the testing cage with a silenced stopwatch. Scored behaviors were described previously (*Kogan et al., 2000*): direct contact with the juvenile including grooming and pawing, sniffing including the anogenital area and mouth and close following (within 1 cm) of the juvenile. After 24 hr, the 10 min test was run again with the previously encountered mouse. Any aggressive encounters observed between animals led to exclusion of the subject mouse from analysis.

## Three-chamber social interaction test

Using a standard three-chamber design (*Moy et al., 2004*), the apparatus consisted of a neutral central zone (18 × 40.5 × 22 cm) connecting two identical compartments (each 19.5 × 40.5 × 22 cm) (*Molas et al., 2017*). Each of the outer compartments housed a caged cylinder (8 cm diameter, 18 cm height; 1 cm between each vertical rod) to allow limited, but direct physical contact between the subject and stimulus animals. Subject mice were placed in the central zone and habituated to an empty apparatus for 5 min then briefly removed to place a juvenile mouse under one of the two caged cylinders while the other caged cylinder remained empty (counterbalanced). The subject mouse was then returned to the apparatus to freely explore all three compartments for 5 min/day for three consecutive days using the same juvenile placed in the same compartment. On day 3, a novel juvenile mouse was placed in the empty caged cylinder. On days 1 and 2, the preference ratio was calculated as: (total social stimulus investigation − total nonsocial stimulus investigation)/(total investigation). On day 3, the preference ratio was calculated as: (total novel stimulus investigation − total familiar stimulus investigation)/(total investigation).

## Tail suspension test

Mice were placed in a rectangular TST apparatus (28 × 28 × 42 cm) and suspended by their tails which were wrapped in red lab tape at around 3/4 the distance from the base. Movement was monitored for 6 min and the last 4 min were scored for immobility behavior (absence of righting attempt).

### Forced swim test

Mice were placed in a plexiglass cylinder (20 cm diameter, 40 cm height) containing 22 ± 1°C water at a depth of 20 cm to prevent them from escaping or touching the bottom. Immobility, measured as floating in the absence of movement except for those necessary to keep the head above water, was measured during the last 4 min of a 6-min session. Following the test, mice were gently dried with a clean paper towel and placed in a fresh cage on top of a heating pad for around 10–15 min after which they were returned to their home cage (*Yankelevitch-Yahav et al., 2015*).

### Fear conditioning

The fear conditioning paradigm was adapted from *Herry et al., 2008*. Animals were not habituated in the testing room to avoid untimely association with auditory cues. Using the ANY-maze fear conditioning system (Ugo Basile SRL), mice were placed in a fear conditioning cage (17 × 17 × 25 cm) in a sound-attenuating box. The paradigm was performed under no light conditions using two different contexts (context A and B). Mice underwent four phases with 24 hr in between each session: habituation, acquisition, auditory recall, and contextual recall. On day 1 (context A), mice were habituated to five 30-s presentations of the CS+ and CS− (white noise) at 80 dB sound pressure level. The inter-cue interval was pseudorandomized and each session with the CS+ or CS− was 10 min. The presentation order of the CS+ and CS− trials were counterbalanced across animals. On day 2 (context A), discriminative fear conditioning was performed by pairing the CS+ with a US (1 s foot shock, 0.75 mA, 5 CS+/ US pairings; intertrial interval: 22–125 s). The onset of the US coincided with the last second of the CS+. On day 3, auditory recall was measured in context B with five presentations of CS+ and CS−. On day 4, the contextual recall was measured in context A for 10 min. ANY-maze software was used to analyze freezing behavior (no movement detected for 1 s), which was scored automatically with an infrared photobeam assay in the fear conditioning cage.

### Object interaction test

The test was adapted from *Molas et al., 2017*. The apparatus consisted of a custom-made white Plexiglass T-shaped maze (three arms, each 9 × 29.5 × 20 cm, connected through a central 9 × 9 cm zone). Mice were placed in the start arm to habituate to the apparatus for 5 min. Following habituation, they were presented with identical inanimate objects located at opposite ends of the T-maze arms for 5 min/day on two consecutive days. On day 3, one of the inanimate objects was replaced with a novel inanimate object placed in the same location (counterbalanced) for 5 min. The preference ratio was calculated from day 3 data as: (total novel stimulus investigation − total familiar stimulus investigation)/(total investigation).

### Marble burying

Fifteen sterilized 1.5-cm glass marbles evenly spaced 2 cm apart in three rows of five were placed in a standard mouse cage with a layer of bedding at a depth of 5–6 cm. A mouse was placed in the cage for 30 min, then returned to its home cage. The number of marbles buried (2/3 of their depth covered with bedding) was counted.

### Grooming

Self-grooming behavior was scored as previously described (*McFarlane et al., 2008*; *Yang et al., 2007*). Mice were habituated for 5 min in an empty mouse cage with no bedding, then grooming behavior was observed for 10 min by an experimenter sitting approximately 2 m from the testing cage. Cumulative time spent grooming during the 10-min session was recorded using a silenced stopwatch.

## Statistical analyses

Results are represented as mean ± standard error of the mean. Statistical significance was set at p < 0.05 and evaluated using paired and unpaired two-tailed Student's *t*-tests and two-way analysis of variance (ANOVA or two-way repeated measures ANOVA with Šidák's post hoc testing for normally distributed data). Mann–Whitney *U*-tests were used for nonparametric data. Analyses were carried out with GraphPad Prism (GraphPad Software). No statistical methods were used to determine sample

sizes prior to the experiments. Sample sizes were comparable to many studies using the similar experimental approaches and animal models. All observed data points were used for statistics.

## Acknowledgements

This work was supported by grants from the National Institutes of Health (R01NS085215 and R01MH130582 to KF, T32 GM107000 and F30MH122146 to AC), the Global Collaborative Research Project of Brain Research Institute, Niigata University (G2905 to KF), and Riccio Neuroscience Fund to KF. The authors thank Ms. Naoe Watanabe for skillful technical assistance. We thank Drs. Veronica Alvarez, Jacqueline N Crawley, Gilles Martin, Motokazu Uchigashima, and David Weaver for comments on an earlier draft of the manuscript.

## Additional information

### Funding

| Funder | Grant reference number | Author |
| --- | --- | --- |
| National Institutes of Health | R01MH130582 | Kensuke Futai |
| National Institutes of Health | R01NS085215 | Kensuke Futai |
| National Institutes of Health | T32 GM107000 | Amy Cheung |
| National Institutes of Health | F30MH122146 | Amy Cheung |

The funders had no role in study design, data collection, and interpretation, or the decision to submit the work for publication.

### Author contributions

Amy Cheung, Conceptualization, Data curation, Software, Formal analysis, Funding acquisition, Validation, Investigation, Visualization, Methodology, Writing – original draft, Writing - review and editing; Kotaro Konno, Data curation, Formal analysis, Validation, Investigation, Visualization, Methodology, Writing – original draft; Yuka Imamura, Data curation, Software, Validation, Investigation, Visualization, Methodology, Writing – original draft; Aya Matsui, Data curation, Software, Supervision, Validation, Visualization, Methodology, Writing – original draft; Manabu Abe, Kenji Sakimura, Resources, Funding acquisition, Investigation, Writing – original draft; Toshikuni Sasaoka, Resources, Data curation, Funding acquisition, Writing – original draft; Takeshi Uemura, Resources, Data curation, Writing – original draft; Masahiko Watanabe, Supervision, Investigation, Visualization, Methodology, Writing – original draft; Kensuke Futai, Conceptualization, Resources, Data curation, Software, Formal analysis, Supervision, Funding acquisition, Validation, Investigation, Visualization, Methodology, Writing – original draft, Project administration, Writing - review and editing

### Author ORCIDs

Amy Cheung  http://orcid.org/0000-0002-4708-0293
Kenji Sakimura  http://orcid.org/0000-0002-8091-8879
Toshikuni Sasaoka  http://orcid.org/0000-0002-7797-4394
Masahiko Watanabe  http://orcid.org/0000-0001-5037-7138
Kensuke Futai  http://orcid.org/0000-0002-3433-3407

### Ethics

All experiments were conducted under approved animal protocols, including protocols #202200005 and #201900338, from the Institutional Animal Care and Use Committee (IACUC) at the University of Massachusetts Chan Medical School.

### Decision letter and Author response

Decision letter https://doi.org/10.7554/eLife.85058.sa1

Author response https://doi.org/10.7554/eLife.85058.sa2

# Additional files

## Supplementary files
• MDAR checklist

## Data availability
Figure 1: published sequencing data set (GSE135132) was used (see below). All data generated or analyzed during this study are included in the manuscript, figures, supporting file, and supporting figures. The meta data set is provided for Figure 1. Previously Published Datasets: Single-cell transcriptomes and whole-brain projections of serotonin neurons in the mouse dorsal and median raphe nuclei: Ren, J. Isakova, A. Friedmann, D. Zeng, J.Grutzner, S. M. Pun, A. Zhao, G. Q. Kolluru, S. S. Wang, R. Lin, R. Li, P. Li, A. Raymond, J. L. Luo, Q.Luo, M. Quake, S. R. Luo, L., 2019, https://elife-sciences.org/articles/49424, GSE135132.

The following previously published dataset was used:

| Author(s) | Year | Dataset title | Dataset URL | Database and Identifier |
|---|---|---|---|---|
| Ren J, Isakova A, Friedmann D, Zeng J, Grutzner SM, Pun A, Zhao GQ, Kolluru SS, Wang R, Lin R, Li P, Li A, Raymond JL, Luo Q, Luo M, Quake SR, Luo L | 2019 | Single-cell transcriptomes and whole-brain projections of serotonin neurons in the mouse dorsal and median raphe nuclei | https://www.ncbi.nlm.nih.gov/geo/query/acc.cgi?acc=GSE135132 | NCBI Gene Expression Omnibus, GSE135132 |

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
