## [Editor Report]

Neurexins control the assembly, maturation, and function of nerve cell synapses, and their genetic loss causes multiple neuropsychiatric diseases, including schizophrenia and autism spectrum disorder (ASD). This manuscript makes an important contribution, by showing convincingly that deletion of all neurexins specifically in serotonergic neurons causes a defect in the survival of serotonergic neurons, in the establishment of serotonergic axonal inputs in various brain regions, in the generation of serotonin release sites, and in serotonin secretion in various brain regions, resulting in ASD-like and depression-related behavioral defects. Thus, not only fast-acting transmitter systems but also modulatory ones depend on neurexin function, and serotonergic signaling contributes to the clinical features of neuropsychiatric disorders caused by neurexin loss. These findings will be interesting to experts in basic neuroscience, psychiatry, and neurology alike.

---

## [Decision Letter]

**Decision letter after peer review:**

[Editors’ note: the authors submitted for reconsideration following the decision after peer review. What follows is the decision letter after the first round of review.]

Thank you for submitting the paper "Neurexins in serotonergic neurons regulate serotonin transmission and complex mouse behaviors" for consideration by *eLife*. Your article has been reviewed by 3 peer reviewers, one of whom is a member of our Board of Reviewing Editors, and the evaluation has been overseen by a Senior Editor. The reviewers have opted to remain anonymous.

We are sorry to say that, after consultation with the reviewers, we have decided that this work will not be considered further for publication by *eLife* at this juncture. As you will see from the detailed reviews below, all three reviewers agree that your study addresses – in an experimentally elegant manner – a very interesting and important neuroscientific problem at the interface between synapse biology and translational psychiatry. However, all three reviewers also identified a major shortcoming that prevents further consideration by *eLife* at this juncture: It remains unclear exactly how serotonergic axons and release sites are altered upon neurexin loss. If this issue can be addressed experimentally, along the lines indicated by the reviewers, *eLife* would be willing to consider a new submission of the manuscript.

Essential Revisions

*Reviewer #1 (Recommendations for the authors):*

Neurexins are adhesion proteins of transmitter-releasing presynaptic compartments that interact with multiple partner proteins to control the assembly, maturation, and function of various types of nerve cell synapses. Neurexin loss-of-function has been linked to multiple neuropsychiatric diseases, including schizophrenia and autism spectrum disorder (ASD), but it is unknown which synapses are affected upon Neurexin loss to cause the characteristic clinical features. It is in this context that the present paper makes an interesting contribution. The manuscript shows that deletion of all Neurexins specifically in serotonergic neurons causes a defect in serotonin release in various brain regions and subtle ASD-like and depression-related behavioural defects. These interesting observations show that not only fast-acting transmitter systems but also modulatory ones depend on Neurexin function, and they are in accord with the notion that defects in serotonergic signalling contribute critically to the clinical features of neuropsychiatric disorders caused by Neurexin loss. However, the present study contains several loose ends that prevent clear-cut conclusions. Most importantly, (i) it remains unclear whether Neurexin loss causes a genuine defect in the serotonin release machinery or simply a reduction of serotonergic axons and release sites, and (ii) the behavioral data require complementary readouts to bolster the still preliminary findings.

The present paper clearly has potential, but appears premature in its current form, even for a short-report-like submission. It documents two interesting initial discoveries, but contains several related loose ends that prevent clear-cut conclusions, as the authors themselves concede in the second and the last paragraphs of their discussion.

1. The voltammetry data in Figure 2 show nicely and convincingly that 5-HT release is reduced in Fev-Cre- Neurexin-TKO dorsal raphe nucleus and hippocampus. It remains unclear, though, whether this phenotype is due to a genuine defect in the 5-HT release machinery or to a reduction of 5-HT axons and release sites in the two brain regions tested, as Figure 3 would indicate. Given the known functions of Neurexins, this is an important distinction that needs to be assessed. This issue could, for instance, be addressed in tissue or neurons of Fev-Cre- Neurexin-TKO and control mice by (i) directly measuring triggered synaptic vesicle fusion in 5-HT terminals using SynaptopHluorin or a related reporter after cell-type specific expression, or by (ii) quantitatively assessing components of the transmitter release machinery or of synaptic vesicle clusters in SERT-positive axons. Furthermore, the data shown in Figure 3 should be complemented by Western blot analyses of SERT levels in hippocampus and dorsal raphe nucleus, which is more sensitive and easier to quantify than immunolabeling in tissue.

2. The behavioral data in Figure 4 indicate subtle ASD-like and depression-related defects in the Fev-Cre- Neurexin-TKO. This is clearly interesting. However, complementary readouts would bolster the still preliminary findings, e.g. the three-chamber test to study sociability and social memory.

*Reviewer #2 (Recommendations for the authors):*

Cheung et al. investigate the role of the Neurexin family of synaptic adhesion proteins in regulating the function of serotonergic neurotransmission. The function of presynaptic Neurexins has been extensively characterized in the regulation of fast synaptic transmission, but whether they also play a role in neuromodulatory systems such as the serotonergic system remains largely unknown. This question is particularly pertinent given that both Neurexins and serotonergic signaling have been linked to the etiology of psychiatric disorders, including schizophrenia and autism spectrum disorders. To address this issue, the authors first demonstrate that serotonergic neurons in the dorsal raphe nucleus express multiple Neurexin isoforms. The authors then delete all Neurexin isoforms specifically from these neurons, and investigate the consequences on 5-HT (serotonin) release, serotonergic innervation, and behavior. They report a substantial reduction in 5-HT release in the dorsal raphe nucleus and hippocampus, accompanied by a modest reduction in serotonin receptor (SERT)-positive fibers as well as alterations in social and depression-like behaviors.

This study addresses an important and timely question. While the molecular mechanisms that govern fast synaptic neurotransmission have been the subject of extensive investigation over several decades, the equivalent mechanisms at neuromodulatory release sites have received far less attention. In recent years this is beginning to change, and the present observation that Neurexins regulate serotonergic neurotransmission provides an interesting contribution to this growing field. The experiments appear thoroughly and carefully conducted, and the manuscript and figures are very well prepared and clearly presented. The behavioral characterization is comprehensive and well executed, even though only a few selective behavioral abnormalities were observed.

However, the conclusions that can be drawn from the present study are somewhat limited in scope, due to the lack of any mechanistic experiments that may begin to explain the observed phenotypes. The study is purely descriptive, and the analysis of serotonergic signaling is essentially limited to the analysis of two relatively general functional parameters, i.e. 5-HT release and density of SERT-positive fibers. Given what is previously known about the role of Neurexin in fast synaptic transmission, many questions arise regarding the mechanisms by which Neurexins may function in a system that is primarily geared toward volume transmission and in which most release sites do not have an associated postsynaptic structure. Importantly, the authors do acknowledge the need for detailed mechanistic studies, and they are careful to present only conclusions that are supported by their data. Nevertheless, additional mechanistic experiments would help to strengthen the notion that Neurexins are important players in serotonergic signaling.

1. The authors state that "Given the predominance of non-junctional specializations, we speculate that Nrxns reside at 5-HT release sites that lack a direct postsynaptic target" (Discussion, p. 7). This is an interesting point and one that could be expanded further, including experimentally. Could it be shown experimentally that Neurexins are present at non-synaptic structures? What extracellular interaction partners might they bind to here?

2. The authors state that their data indicate "that the primary role of Nrxns in the 5-HT system is the formation of functional components important for 5-HT release" (Discussion, p. 7). This too could be tested experimentally, e.g. through immunohistochemical oder ultrastructural analysis of presynaptic terminals.

3. What are the consequences of conditional single deletion of Nrxn1, 2 or 3 on the key phenotypes?

4. Please state whether all experiments were conducted blind to genotype. This is mentioned in some but not in other sections, leaving the impression that not all experiments were conducted blind to genotype – is this true? If so, it should be stated explicitly.

5. Were any controls conducted with Fev-Cre mice alone to exclude an effect of Cre expression on serotonergic synaptic transmission and behavior, or have these controls been previously published? If not, please discuss the possibility that the observed changes may result from Fev-Cre expression rather than from the deletion of Neurexins.

6. For the behavioral experiments, mice were used at age 8 weeks or older. Is this also true for the other experiments? Please state the age range used for each experiment.

7. Please state the genetic background for the mice.

*Reviewer #3 (Recommendations for the authors):*

This manuscript is the first to report effects of genetic deletion of neurexins from serotonergic neurons. The authors used a nice combination of fast-scan cyclic voltammetry with electrical stimulation to assess serotonin release, immunofluorescence for serotonin transporter, and a wide range of behavioral assays. Loss of neurexins reduced serotonin release and serotonin transporter immunoreactivity in the dorsal raphe nucleus and dorsal hippocampus. The conditional knockout mice showed reduced social interaction and increased depressive-like behavior in the forced swim test. These data constitute strong evidence for a function of neurexins in serotonergic neurons in neurotransmission and in a subset of behaviors. Overall, the conclusions are well supported by the data. This analysis of the role of neurexins in serotonergic neurons is an important contribution to the field.

1. One aspect which would benefit from further analysis is a more in-depth study of exactly how serotonergic axons and release sites are altered upon loss of neurexins. As the authors discuss, the observed difference in SERT immunofluorescence could reflect a difference in axon arbors or a difference in SERT overall expression or local aggregation. Since the conditional KO mice include a tdTomato reporter, is there enough signal from the tdTomato (alone or amplified by immunostaining) to determine whether loss of neurexins affects the axon arbors? Overall expression of SERT could be assessed by Western blot. It would also be interesting to assess the localization of active zone proteins in the serotonergic axons. Co-labeling of bassoon, ELKS, or RIM with SERT could be assessed by a super-resolution or expansion microscopy approach. Co-labeling of SERT with VGlut3 might also be informative. I do not expect all of these additional experiments to be performed but some further information would strengthen the manuscript.

2. While I follow the reason for studying the DRN, it is puzzling to me why the authors focused additionally on the dorsal hippocampus rather than a region with stronger serotonergic innervation. The effect of fluoxetine on the FSCV signal is substantial in DRN but weak in the hippocampus raising some question about specificity of the signal for serotonin.

3. There is a related publication that was not cited. Seigneur et al. (2021; Molecular Psychiatry; https://doi.org/10.1038/s41380-021-01187-x) found that Cbln-2 functions in dorsal raphe serotonergic neurons in regulating serotonin release and behavior. Since the only known signaling mechanism for Cbln-2 is through a neurexin-Cbln-GluD complex, this indirectly implicates neurexins. It would be a valuable discussion point to compare the current results with those of Seigneur et al. (2021).

4. It may be best to refer to the control mice as 'control' or 'Con' rather than 'WT' since these were Cre-negative littermates and Fev/RFP mice, thus mostly not strictly WT.

5. There seems to be very low sensitivity in the Q-PCR assay in Figure 1D, as only 2 or 4 of the 23 control cells showed signal for Nrxn2 and Nrxn3, respectively, although a greater fraction of cells express these genes (panel C). Pooling cells may be preferable. The more effective detection of Nrxn1 in control cells confirms its deletion with Fev-Cre, so it is likely that Nrxn 2 and 3 were also deleted.

[Editors’ note: further revisions were suggested prior to acceptance, as described below.]

Thank you for resubmitting your work entitled "Neurexins in serotonergic neurons regulate serotonin transmission and complex mouse behaviors" for further consideration by *eLife*. Your revised article has been evaluated by Gary Westbrook (Senior Editor) and a Reviewing Editor.

The manuscript has been improved but there are three remaining issues that need to be addressed, and a suggestion, as outlined below:

1. To demonstrate the involvement of Nrxns in 5HT release in the hippocampus, the authors change the statistical analysis in Figures 2F and 2J from a two-way repeated measure ANOVA to a paired t-test. This new statistical analysis now supposedly shows that fluoxetine treatment increases 5HT transients in control mice but not in TKO mice in the hippocampus, leading to the conclusion that "the lack of statistical significance in the TKO group is due to greatly reduced 5-HT release in TKO slices that reaches the limitation of FSCV sensitivity". However, there is no justification for changing this statistical analysis. The original analysis, which used a two-way repeated measures ANOVA, was fully correct and convincingly demonstrated a lack of a genotype effect (genotype main effect, F1,9 = 0.01351, p = 0.91; drug x genotype interaction, F1,9 = 0.09308, p = 0.7672). The paired t-test now conducted for the revised manuscript is an inappropriate statistical analysis, since there is no statistical comparison of genotypes, only of drug effect within each genotype, making it impossible to make any statistical claims on a genotype effect. This analysis must be corrected back to the original version, i.e. the two-way repeated measures ANOVA, to avoid falsifying the conclusions.

2. The authors must provide information on the age of the mice used for the RT-PCR and voltammetry experiments (Figures 1 and 2). The only information on the age of mice is given for data in Figures 3 and 4, it appears. To facilitate finding this information, the authors must add a summary of the ages of the mice for all experiments to the 'Animals' section of the Methods part. Given that Nrxns are differentially involved at different time points of synapse development and function, it is important to know at which developmental time points analyses were performed.

3. As regards additional behavioral experiments, the reviewers acknowledge that the authors did the three-chamber test to assess social interaction, but did not obtain further evidence for ASD-like behavior in the mutants. This warrants a more conservative, differentiated, discussion of the role of defects in the serotonergic system in neuropsychiatric conditions caused by Nrxn loss.

4. The reviewers suggest including the effect on neuron survival in the title as it is a rather surprising and important finding.

---

## [Author Response]

[Editors’ note: the authors resubmitted a revised version of the paper for consideration. What follows is the authors’ response to the first round of review.]

Essential RevisionsReviewer #1 (Recommendations for the authors):Neurexins are adhesion proteins of transmitter-releasing presynaptic compartments that interact with multiple partner proteins to control the assembly, maturation, and function of various types of nerve cell synapses. Neurexin loss-of-function has been linked to multiple neuropsychiatric diseases, including schizophrenia and autism spectrum disorder (ASD), but it is unknown which synapses are affected upon Neurexin loss to cause the characteristic clinical features. It is in this context that the present paper makes an interesting contribution. The manuscript shows that deletion of all Neurexins specifically in serotonergic neurons causes a defect in serotonin release in various brain regions and subtle ASD-like and depression-related behavioural defects. These interesting observations show that not only fast-acting transmitter systems but also modulatory ones depend on Neurexin function, and they are in accord with the notion that defects in serotonergic signalling contribute critically to the clinical features of neuropsychiatric disorders caused by Neurexin loss. However, the present study contains several loose ends that prevent clear-cut conclusions. Most importantly, (i) it remains unclear whether Neurexin loss causes a genuine defect in the serotonin release machinery or simply a reduction of serotonergic axons and release sites, and (ii) the behavioral data require complementary readouts to bolster the still preliminary findings.

We thank the reviewer for this comment. In the revised manuscript, we address (i) by exploring deficits in serotonin release machinery, and (ii) by reporting the results of an additional behavioral approach (see below) and discussing the mild behavioral effects.

The present paper clearly has potential, but appears premature in its current form, even for a short-report-like submission. It documents two interesting initial discoveries, but contains several related loose ends that prevent clear-cut conclusions, as the authors themselves concede in the second and the last paragraphs of their discussion.

We appreciate the Reviewer’s interest in our initial discoveries. The updated manuscript provides more clear-cut conclusions. Notably, serotonergic Nrxns are important for 5-HT neuron survival and release machinery.

1. The voltammetry data in Figure 2 show nicely and convincingly that 5-HT release is reduced in Fev-Cre- Neurexin-TKO dorsal raphe nucleus and hippocampus. It remains unclear, though, whether this phenotype is due to a genuine defect in the 5-HT release machinery or to a reduction of 5-HT axons and release sites in the two brain regions tested, as Figure 3 would indicate. Given the known functions of Neurexins, this is an important distinction that needs to be assessed. This issue could, for instance, be addressed in tissue or neurons of Fev-Cre- Neurexin-TKO and control mice by (i) directly measuring triggered synaptic vesicle fusion in 5-HT terminals using SynaptopHluorin or a related reporter after cell-type specific expression, or by (ii) quantitatively assessing components of the transmitter release machinery or of synaptic vesicle clusters in SERT-positive axons. Furthermore, the data shown in Figure 3 should be complemented by Western blot analyses of SERT levels in hippocampus and dorsal raphe nucleus, which is more sensitive and easier to quantify than immunolabeling in tissue.

We appreciate this comment. Our collaborators, Drs. Konno and Watanabe, are experts in high-resolution immunohistochemistry imaging and performed confocal imaging against control and TKO tissues given our limited experience with super-resolution microscopy. We first confirmed the expression of active zone proteins, RIM1/2, in SERT-positive fibers in the dorsal hippocampal CA1 region in ultra-thin (100 nm) sections (Figure 3—figure supplement 2) and then measured the density of RIM1/2 in SERT/RFP-positive fibers (Figure 3N and 3O). We believe that using ultra-thin sections as a confirmational experiment allowed us to use our existing resource of confocal microscopy to quantify RIM1/2 proteins localized in serotonergic terminals. Importantly, we found that RIM1/2 density is significantly reduced in Fev/RFP/NrxnTKO mice compared with control mice. This imaging study suggests that Nrxn TKO in 5-HT fibers reduced active zone density and therefore 5-HT release sites.

2. The behavioral data in Figure 4 indicate subtle ASD-like and depression-related defects in the Fev-Cre- Neurexin-TKO. This is clearly interesting. However, complementary readouts would bolster the still preliminary findings, e.g. the three-chamber test to study sociability and social memory.

We performed a three-chamber social interaction test (3CST) as presented in Figure 4—figure supplement 2. Fev/RFP/NrxnTKO mice showed similar performance to Cntl mice. However, Cntl and Fev/RFP/NrxnTKO mice showed different investigation behavior on Day 1. There were no differences in exploration time between groups which contrasts the sociability deficits observed in the direct social interaction test. We consider that the different results of the direct social interaction test and 3CST can be due to two possibilities: (i) 3CST limits direct interaction with a stimulus mouse and it is possible that the mode and novelty of social interaction are critical to Fev/RFP/NrxnTKO mice, as there were genotype differences observed on initial encounter in both experiments, and (ii) the mild behavioral deficit in Fev/RFP/NrxnTKO mice can be due to intact serotonergic circuitry mediated by Cre-negative 5-HT neurons (which constitute approximately 40% of the total 5-HT neurons, Figure 3M). We state these possibilities in the Discussion section (page 8).

Reviewer #2 (Recommendations for the authors):Cheung et al. investigate the role of the Neurexin family of synaptic adhesion proteins in regulating the function of serotonergic neurotransmission. The function of presynaptic Neurexins has been extensively characterized in the regulation of fast synaptic transmission, but whether they also play a role in neuromodulatory systems such as the serotonergic system remains largely unknown. This question is particularly pertinent given that both Neurexins and serotonergic signaling have been linked to the etiology of psychiatric disorders, including schizophrenia and autism spectrum disorders. To address this issue, the authors first demonstrate that serotonergic neurons in the dorsal raphe nucleus express multiple Neurexin isoforms. The authors then delete all Neurexin isoforms specifically from these neurons, and investigate the consequences on 5-HT (serotonin) release, serotonergic innervation, and behavior. They report a substantial reduction in 5-HT release in the dorsal raphe nucleus and hippocampus, accompanied by a modest reduction in serotonin receptor (SERT)-positive fibers as well as alterations in social and depression-like behaviors.This study addresses an important and timely question. While the molecular mechanisms that govern fast synaptic neurotransmission have been the subject of extensive investigation over several decades, the equivalent mechanisms at neuromodulatory release sites have received far less attention. In recent years this is beginning to change, and the present observation that Neurexins regulate serotonergic neurotransmission provides an interesting contribution to this growing field. The experiments appear thoroughly and carefully conducted, and the manuscript and figures are very well prepared and clearly presented. The behavioral characterization is comprehensive and well executed, even though only a few selective behavioral abnormalities were observed.However, the conclusions that can be drawn from the present study are somewhat limited in scope, due to the lack of any mechanistic experiments that may begin to explain the observed phenotypes. The study is purely descriptive, and the analysis of serotonergic signaling is essentially limited to the analysis of two relatively general functional parameters, i.e. 5-HT release and density of SERT-positive fibers. Given what is previously known about the role of Neurexin in fast synaptic transmission, many questions arise regarding the mechanisms by which Neurexins may function in a system that is primarily geared toward volume transmission and in which most release sites do not have an associated postsynaptic structure. Importantly, the authors do acknowledge the need for detailed mechanistic studies, and they are careful to present only conclusions that are supported by their data. Nevertheless, additional mechanistic experiments would help to strengthen the notion that Neurexins are important players in serotonergic signaling.

We appreciate this Reviewer’s supportive comments and have updated our revised manuscript with more mechanistic experiments to strengthen the role of Nrxns in serotonergic signaling, as discussed above.

1. The authors state that "Given the predominance of non-junctional specializations, we speculate that Nrxns reside at 5-HT release sites that lack a direct postsynaptic target" (Discussion, p. 7). This is an interesting point and one that could be expanded further, including experimentally. Could it be shown experimentally that Neurexins are present at non-synaptic structures? What extracellular interaction partners might they bind to here?

It is certainly interesting to address the expression of postsynaptic targets in Fev/RFP/NrxnTKO mouse brain. Unfortunately, we have had issues with mouse breeding related to the construction of a new research building next to our animal facility. So far, we have a limited number of animals and have established reliable imaging setting only for triple staining RFP, RIM1/2, and SERT proteins. While we have continuously tried to address this question by establishing multi-color imaging protocols that stain RFP (or SERT), active zone marker (RIM1/2 and others), and postsynaptic targets, including Neuroligin 1, 2 and 3, we haven’t established the staining conditions yet.

The staining of Nrxns has been a significant challenge in our field. We have tested 5 commercially available Nrxn antibodies, however, none of them were validated by our KO tissue.

2. The authors state that their data indicate "that the primary role of Nrxns in the 5-HT system is the formation of functional components important for 5-HT release" (Discussion, p. 7). This too could be tested experimentally, e.g. through immunohistochemical oder ultrastructural analysis of presynaptic terminals.

We appreciate this suggestion. We performed high-resolution immunohistochemistry imaging against control and TKO tissues. We first confirmed the expression of the active zone marker, RIM1/2, in SERT-positive fibers in ultra-thin (100 nm) sections, and then performed triple staining against RFP, SERT, and RIM1/2 in ultra-thin sections prepared from Fev/RFP (Cntl) and Fev/RFP/NrxnTKO mice. Importantly, NrxnTKO in 5-HT neurons reduced RIM1/2 density in Fev/RFP/NrxnTKO mice. These results are presented in Figure 3N, 3O, and Figure 3—figure supplement 2.

3. What are the consequences of conditional single deletion of Nrxn1, 2 or 3 on the key phenotypes?

We agree that it is important to find Nrxn isoform(s) responsible for the phenotypes we found in this manuscript. However, this manuscript focuses on the general role of Nrxns in the 5-HT signaling. This comment requires significant time and budget to generate single or double Nrxn KO mouse lines. Our school has begun construction for a new research building right next to our animal facility which has caused considerable issues with mouse breeding. Therefore, we would like to address this comment as a future project.

4. Please state whether all experiments were conducted blind to genotype. This is mentioned in some but not in other sections, leaving the impression that not all experiments were conducted blind to genotype – is this true? If so, it should be stated explicitly.

Addressed. All experiments and data analysis were performed in a blind manner and we refer to this in the Materials and methods section.

5. Were any controls conducted with Fev-Cre mice alone to exclude an effect of Cre expression on serotonergic synaptic transmission and behavior, or have these controls been previously published? If not, please discuss the possibility that the observed changes may result from Fev-Cre expression rather than from the deletion of Neurexins.

Fev-Cre mice were used as controls in experiments that are specified in the manuscript, including for RFP-positive cell density and fiber analysis and validation of the Fev/RFP/NrxnTKO mouse line. The effect of Cre expression on 5-HT function has not been previously published and we have discussed the unlikely possibility that the observed changes may result from Fev-Cre expression rather than from the deletion of Nrxns in the Discussion section (page 8).

6. For the behavioral experiments, mice were used at age 8 weeks or older. Is this also true for the other experiments? Please state the age range used for each experiment.

We now state clearly the age range of mice for each experiment in the Materials and methods section.

7. Please state the genetic background for the mice.

Fev-Cre and TdTomato reporter lines were backcrossed with C57BL6 line for at least 10 generations. The Nrxn floxed line was generated using C57BL6 ES cells (T. Uemura et al., 2022). We have included these details in the Animals section in the Materials and methods section.

Reviewer #3 (Recommendations for the authors):This manuscript is the first to report effects of genetic deletion of neurexins from serotonergic neurons. The authors used a nice combination of fast-scan cyclic voltammetry with electrical stimulation to assess serotonin release, immunofluorescence for serotonin transporter, and a wide range of behavioral assays. Loss of neurexins reduced serotonin release and serotonin transporter immunoreactivity in the dorsal raphe nucleus and dorsal hippocampus. The conditional knockout mice showed reduced social interaction and increased depressive-like behavior in the forced swim test. These data constitute strong evidence for a function of neurexins in serotonergic neurons in neurotransmission and in a subset of behaviors. Overall, the conclusions are well supported by the data. This analysis of the role of neurexins in serotonergic neurons is an important contribution to the field.

We are grateful for this Reviewer’s comment about our manuscript.

1. One aspect which would benefit from further analysis is a more in-depth study of exactly how serotonergic axons and release sites are altered upon loss of neurexins. As the authors discuss, the observed difference in SERT immunofluorescence could reflect a difference in axon arbors or a difference in SERT overall expression or local aggregation. Since the conditional KO mice include a tdTomato reporter, is there enough signal from the tdTomato (alone or amplified by immunostaining) to determine whether loss of neurexins affects the axon arbors? Overall expression of SERT could be assessed by Western blot. It would also be interesting to assess the localization of active zone proteins in the serotonergic axons. Co-labeling of bassoon, ELKS, or RIM with SERT could be assessed by a super-resolution or expansion microscopy approach. Co-labeling of SERT with VGlut3 might also be informative. I do not expect all of these additional experiments to be performed but some further information would strengthen the manuscript.

We appreciate this comment. First, we performed western blotting against SERT and found that NrxnTKO in 5-HT neurons reduced SERT expression in the brainstem and hippocampus, but not in the midbrain. These results are incorporated in Figure 3—figure supplement 1. Next, we performed confocal imaging labeling RFP, SERT, and RIM1/2 (active zone marker) in Fev/RFP/NrxnTKO and Fev/RFP (control) mice. Because our department is not equipped to perform super-resolution microscopy, we overcame the z-axis resolution issue by imaging ultra-thin sections (100 nm) and confirmed RIM1/2 expression in SERT fibers. Our new findings strongly support that Nrxn TKO in 5-HT neurons reduces the number of release sites in serotonergic terminals. These results are included in Figures 3N and 3O, and Figure 3—figure supplement 2.

2. While I follow the reason for studying the DRN, it is puzzling to me why the authors focused additionally on the dorsal hippocampus rather than a region with stronger serotonergic innervation. The effect of fluoxetine on the FSCV signal is substantial in DRN but weak in the hippocampus raising some question about specificity of the signal for serotonin.

We focused on the hippocampal circuit because first, we observed reduced SERT expression in the hippocampus, and second, we observed abnormal direct social interaction which can be regulated by the dorsal hippocampal circuit (Hitti & Siegelbaum, 2014). We apologize that we chose the wrong statistical method to quantify the effect of fluoxetine to confirm 5-HT release in FSCV. In the initial manuscript, we chose a two-way repeated measures ANOVA for this experiment. However, since we are comparing FSCV transients before and after fluoxetine treatment within the genotype, a paired t-test for each genotype should be applied. The new statistical analysis using paired t-test demonstrates statistical significance in control but not Fev/RFP/NrxnTKO hippocampus. We consider that the reason for the lack of statistical significance in the TKO group is due to greatly reduced 5-HT release in TKO slices that reaches the limitation of FSCV sensitivity.

3. There is a related publication that was not cited. Seigneur et al. (2021; Molecular Psychiatry; https://doi.org/10.1038/s41380-021-01187-x) found that Cbln-2 functions in dorsal raphe serotonergic neurons in regulating serotonin release and behavior. Since the only known signaling mechanism for Cbln-2 is through a neurexin-Cbln-GluD complex, this indirectly implicates neurexins. It would be a valuable discussion point to compare the current results with those of Seigneur et al. (2021).

We appreciate this suggestion. We included this paper in the Discussion section (page 7).

4. It may be best to refer to the control mice as 'control' or 'Con' rather than 'WT' since these were Cre-negative littermates and Fev/RFP mice, thus mostly not strictly WT.

We appreciate this suggestion. We switched our abbreviation from WT to control (Cntl).

5. There seems to be very low sensitivity in the Q-PCR assay in Figure 1D, as only 2 or 4 of the 23 control cells showed signal for Nrxn2 and Nrxn3, respectively, although a greater fraction of cells express these genes (panel C). Pooling cells may be preferable. The more effective detection of Nrxn1 in control cells confirms its deletion with Fev-Cre, so it is likely that Nrxn 2 and 3 were also deleted.

We appreciate this suggestion. We performed digital PCR against pooled single-cell cDNA libraries and confirmed that pooled Nrxn TKO in 5-HT neurons express significantly lower levels of Nrxn2 and Nrxn3 genes compared with that of control neurons. These results are included in Figure 1—figure supplement 1.

[Editors’ note: what follows is the authors’ response to the second round of review.]

The manuscript has been improved but there are three remaining issues that need to be addressed, and a suggestion, as outlined below:1. To demonstrate the involvement of Nrxns in 5HT release in the hippocampus, the authors change the statistical analysis in Figures 2F and 2J from a two-way repeated measure ANOVA to a paired t-test. This new statistical analysis now supposedly shows that fluoxetine treatment increases 5HT transients in control mice but not in TKO mice in the hippocampus, leading to the conclusion that "the lack of statistical significance in the TKO group is due to greatly reduced 5-HT release in TKO slices that reaches the limitation of FSCV sensitivity". However, there is no justification for changing this statistical analysis. The original analysis, which used a two-way repeated measures ANOVA, was fully correct and convincingly demonstrated a lack of a genotype effect (genotype main effect, F1,9 = 0.01351, p = 0.91; drug x genotype interaction, F1,9 = 0.09308, p = 0.7672). The paired t-test now conducted for the revised manuscript is an inappropriate statistical analysis, since there is no statistical comparison of genotypes, only of drug effect within each genotype, making it impossible to make any statistical claims on a genotype effect. This analysis must be corrected back to the original version, i.e. the two-way repeated measures ANOVA, to avoid falsifying the conclusions.

We appreciate this comment. As suggested by the reviewers, we changed our statistical method for the FLX experiment in Figures 2F and 2J back to two-way repeated measures ANOVA test.

2. The authors must provide information on the age of the mice used for the RT-PCR and voltammetry experiments (Figures 1 and 2). The only information on the age of mice is given for data in Figures 3 and 4, it appears. To facilitate finding this information, the authors must add a summary of the ages of the mice for all experiments to the 'Animals' section of the Methods part. Given that Nrxns are differentially involved at different time points of synapse development and function, it is important to know at which developmental time points analyses were performed.

We apologize that we did not include the age information of animals that were tested in RT-qPCR and voltammetry experiments. The revised manuscript includes the ages of animals in these experiments in the Materials and methods section, including the Animals and other sections.

3. As regards additional behavioral experiments, the reviewers acknowledge that the authors did the three-chamber test to assess social interaction, but did not obtain further evidence for ASD-like behavior in the mutants. This warrants a more conservative, differentiated, discussion of the role of defects in the serotonergic system in neuropsychiatric conditions caused by Nrxn loss.

We agree with the reviewers’ comment that our new behavior data doesn’t provide further evidence for ASD-like behavior in the mutant mice. Our revised manuscript includes conservative statements in the Discussion section.

4. The reviewers suggest including the effect on neuron survival in the title as it is a rather surprising and important finding.

Addressed. The revised title is “Neurexins in serotonergic neurons regulate neuronal survival, serotonin transmission, and complex mouse behaviors”.